# Inhibitors of Rho kinases (ROCK) induce multiple mitotic defects and synthetic lethality in BRCA2-deficient cells

Julieta Martino[1†], Sebastián Omar Siri[1†], Nicolás Luis Calzetta[1], Natalia Soledad Paviolo[1], Cintia Garro[2,3], Maria F Pansa[2‡], Sofía Carbajosa[2,3], Aaron C Brown[4], José Luis Bocco[2], Israel Gloger[5], Gerard Drewes[5], Kevin P Madauss[6], Gastón Soria[2,3*], Vanesa Gottifredi[1*]

[1]Fundación Instituto Leloir-CONICET, Buenos Aires, Argentina; [2]Centro de Investigaciones en Bioquímica Clínica e Inmunología, CIBICI-CONICET, Departamento de Bioquímica Clínica, Facultad de Ciencias Químicas, Universidad Nacional de Córdoba, Córdoba, Argentina; [3]OncoPrecision, Córdoba, Argentina; [4]Center for Molecular Medicine, Maine Medical Center Research Institute, Scarborough, United States; [5]GlaxoSmithKline-Trust in Science, Global Health R&D, Stevenage, United Kingdom; [6]GlaxoSmithKline-Trust in Science, Global Health R&D, Upper Providence, United States

**\*For correspondence:**
gsoria29@gmail.com (GS);
vgottifredi@leloir.org.ar (VG)

[†]These authors contributed equally to this work

**Present address:**
[‡]GlaxoSmithKline-Trust in Science, Global Health R&D, Upper Providence, United States

**Abstract** The trapping of Poly-ADP-ribose polymerase (PARP) on DNA caused by PARP inhibitors (PARPi) triggers acute DNA replication stress and synthetic lethality (SL) in BRCA2-deficient cells. Hence, DNA damage is accepted as a prerequisite for SL in BRCA2-deficient cells. In contrast, here we show that inhibiting ROCK in BRCA2-deficient cells triggers SL independently from acute replication stress. Such SL is preceded by polyploidy and binucleation resulting from cytokinesis failure. Such initial mitosis abnormalities are followed by other M phase defects, including anaphase bridges and abnormal mitotic figures associated with multipolar spindles, supernumerary centrosomes and multinucleation. SL was also triggered by inhibiting Citron Rho-interacting kinase, another enzyme that, similarly to ROCK, regulates cytokinesis. Together, these observations demonstrate that cytokinesis failure triggers mitotic abnormalities and SL in BRCA2-deficient cells. Furthermore, the prevention of mitotic entry by depletion of Early mitotic inhibitor 1 (EMI1) augmented the survival of BRCA2-deficient cells treated with ROCK inhibitors, thus reinforcing the association between M phase and cell death in BRCA2-deficient cells. This novel SL differs from the one triggered by PARPi and uncovers mitosis as an Achilles heel of BRCA2-deficient cells.

## Editor's evaluation

This paper reports the fundamental discovery that BRCA2-deficient cells are highly sensitive to the inhibition or depletion of Rho-kinases (ROCK), known to regulate actin cytoskeleton dynamics. This observed synthetic lethality between ROCK and BRCA2 is suggested to be independent of acute replication stress, is outside of the cellular S phase and may represent a promising new synthetic lethality target for the treatment of BRCA2-deficient tumors.

## Introduction

Hereditary breast and ovarian cancer (HBOC) is an autosomal dominant disease that accounts for 5–10% of breast (*Krainer et al., 1997*; *Langston et al., 1996*) and 15% of ovarian cancer cases

(*Pal et al., 2005*; *Zhang et al., 2011*). HBOC is primarily caused by mutations in the breast cancer susceptibility genes BRCA1 and BRCA2 (*Futreal et al., 1994*; *Miki et al., 1994*; *Wooster et al., 1995*). BRCA1 and BRCA2 are DNA repair genes, and their protein products regulate homologous recombination (HR), a repair pathway that is recruited to highly toxic DNA double-strand breaks (DSBs; *Prakash et al., 2015*). Additionally, BRCA1 and BRCA2 are essential for DNA replication events, including replication fork protection, reversal, restart and gap-filling (*Cong and Cantor, 2022*; *Cong et al., 2021*; *Panzarino et al., 2021*; *Ray Chaudhuri et al., 2016*; *Schlacher et al., 2012*). BRCA1- and BRCA2-deficient cells exhibit structural chromosome abnormalities and are highly sensitive to DNA-damaging agents (*Moynahan et al., 2001*; *Patel et al., 1998*; *Yu et al., 2000*). Additionally, BRCA-deficient cells exhibit translocations, large deletions and chromosome fusions (*Moynahan et al., 2001*; *Yu et al., 2000*). This chromosome instability underlies the tumorigenicity of BRCA-deficient tumors and underscores the critical tumor suppressor function of BRCA genes in cells.

Mutations in BRCA genes are highly penetrant, and their carriers have a high risk of developing early-onset breast and ovarian cancer (*Antoniou et al., 2003*; *King et al., 2003*). Carriers of BRCA mutations are also at an increased risk of developing other malignancies, including pancreatic and prostate cancers and melanoma (*Cavanagh and Rogers, 2015*; *Gumaste et al., 2015*). BRCA-mutation carriers whose mutations are detected before cancer onset are suggested to undergo highly invasive surgeries such as salpingo-oophorectomy and mastectomy. The standard of care for BRCA-mutation carriers with tumors is similar to the approach used for patients with sporadic tumors, except for some types of BRCA-deficient tumors, which might be more sensitive to platinum-based therapies (*Vencken et al., 2011*; *Yang et al., 2011*). Unfortunately, chemotherapy resistance to platinum agents is common and alternative therapies are most needed for these patients.

One group of alternative therapeutic agents that are clinically available is poly-ADP-ribose polymerase (PARP) inhibitors which are highly effective in killing BRCA-deficient cells (*Bryant et al., 2005*; *Farmer et al., 2005*; *McCabe et al., 2006*) and several PARP inhibitors (PARPi) have been approved for clinical use. The synthetic lethality (SL) observed between BRCA deficiency, and PARPi is due to the ability of PARPi to physically trap PARP on DNA (*Murai et al., 2014*; *Murai et al., 2012*). PARP trapping causes the accumulation of DNA replication intermediates, such as gaps, which must be handled by BRCA proteins to protect DNA integrity (*Taglialatela et al., 2021*; *Tirman et al., 2021*). Additionally, some DNA structures that derive from the encounter of replication forks with PARP-bound DNA may require HR-mediated repair, a mechanism impaired in BRCA1- and BRCA2-deficient cells (*Prakash et al., 2015*). While the impaired DNA damage response of BRCA-deficient cells to PARPi leads to cell death, resistance to PARPi is also observed in the clinic (*Barber et al., 2013*). Molecular mechanisms of resistance to PARPi include, but are not limited to, secondary mutations that restore HR function, increased drug efflux, and decreased PARP trapping (*D'Andrea, 2018*; *Noordermeer and van Attikum, 2019*).

As mentioned above, although BRCA proteins were initially studied based on their roles in HR, we currently know that BRCA1 and BRCA2 have pleiotropic functions, performing functions outside canonical HR (*Petsalaki and Zachos, 2020*). Thus, it is likely that multiple targets not restricted to HR could be exploited for SL therapeutic approaches. This concept has been corroborated for BRCA1 deficiency in a phenotypic screening in which we tested BRCA-deficient cells for SL against the kinase inhibitor library PKIS2 (*Carbajosa et al., 2019*). Our findings unveiled that BRCA1-deficient cells have increased sensitivity to Polo-like kinase 1 (PLK1) inhibitors and that this sensitivity does not require excess DNA damage caused by external agents.

In this study, we present findings indicating that BRCA2-deficient cells are highly sensitive to the inhibition or depletion of Rho-kinases (ROCK), which regulate actin cytoskeleton dynamics. Unlike PARPi, ROCK inhibitors (ROCKi) did not induce acute replication stress in BRCA2-deficient cells but instead triggered mitotic defects including cytokinesis failure, polyploidy, aberrant multipolar spindles and centrosome amplification. Remarkably, SL-induction was also observed after inhibition of Citron Rho-interacting kinase (CITK), an enzyme that regulates cytokinesis at the level of mitotic furrow cleavage, indicating that cytokinesis failure is the likely trigger of this novel SL interaction. Moreover, preventing mitotic entry via depletion of Early mitotic inhibitor 1 (EMI1), abrogated ROCKi-induced BRCA2-deficient cell death. In conclusion, while the accumulation of DNA damage in S phase is required for PARPi-mediated cell death (*Ray Chaudhuri et al., 2016*; *Schoonen et al., 2017*), our

findings highlight that BRCA2-deficient cells bear additional vulnerabilities outside S phase that could represent promising new SL targets.

## Results

### BRCA2-deficient cells are sensitive to ROCK inhibition

In a previous work (*Carbajosa et al., 2019*), we developed a phenotypic survival screening method to evaluate the differential sensitivity of BRCA1-deficient cells against 680 ATP-competitive kinase inhibitors provided by GlaxoSmithKline (*Drewry et al., 2017*; *Elkins et al., 2016*). Briefly, the screening was performed using HCT116$^{p21-/-}$ cell lines in which BRCA1 or BRCA2 were stably downregulated using shRNA (*Figure 1A*). This strategy allowed a comparison of BRCA-proficient vs BRCA-deficient cell lines on an isogenic background. In addition, HCT116$^{p21-/-}$ cells are easy to grow and tolerate low seeding densities compatible with long-term (i.e. 6 days) survival analysis. Additionally, we used a p21 knockout background, which attenuates the cell cycle arrest that otherwise would mask the cytotoxic phenotypes during the screening time frame.

In this work, we analyzed the screening results of the BRCA2-deficient cell population. BRCA2 depletion by shRNA in HCT116$^{p21-/-}$ cells was sufficient to trigger increased sensitivity to olaparib (*Figure 1B–C*). For the analysis, we focused on compounds that induced SL exclusively in the BRCA2-deficient population and were not toxic to control samples or BRCA1-deficient cells (*Figure 1D*). Interestingly, BRCA2-deficient cells showed remarkable sensitivity to three inhibitors of ROCK kinases (ROCK) (*Figure 1E* and *Table 1*). The selective activity of all ROCK inhibitors was further validated at a higher dose (*Table 1*) and in a dose-response curve for the three most potent ones (*Figure 1F*).

To test the sensitivity of BRCA2-depleted cells to ROCK inhibition, we took advantage of three commercially available ROCK inhibitors (ROCKi). Two of them are fasudil and ripasudil, which are approved for diseases other than cancer (*Garnock-Jones, 2014*; *Shi and Wei, 2013*). Both are ATP-competitive inhibitors targeting ROCK1 and ROCK2 (*Nakagawa et al., 1996*). In addition, we used the inhibitor SR 3677 dihydrochloride, which is a newer ROCK inhibitor that has interesting advantages such as a low IC50 and high potency in biochemical and cell-based assays as well as high selectivity for ROCK (*Feng et al., 2008*). We performed survival assays with fasudil in several cellular models of BRCA2 deficiency, including the HCT116$^{p21-/-}$ cell line used in the screening (*Figure 2A*). We also tested survival in DLD-1/DLD-1$^{BRCA2-/-}$ paired cell lines, which are BRCA2 knockout (*Figure 2B*) and the PEO4/PEO1, V-C8 #13 /V-C8 paired cell lines (see description of cell lines in the methods section - *Figure 2C–D*). SL was observed in all BRCA2-deficient cell line models following fasudil treatment (*Figure 2A–D*). Cell death was confirmed using SYTOX green, a dye that only enters cells when cellular membranes have been compromised (*Figure 2E*) and in clonogenic survival assays (*Figure 2—figure supplement 1*). Similar differences between control and BRCA2-deficient counterparts were observed with ripasudil and SR 3677 dihydrochloride, two other ROCKi (*Figure 2—figure supplement 2A–C*). In contrast, the BRCA1-deficient cell line HCC1937 (*Tomlinson et al., 1998*), which is sensitive to olaparib (*Figure 2—figure supplement 2D*), did not exhibit increased sensitivity to fasudil or ripasudil compared to the complemented HCC1937$^{BRCA1}$ cell line (*Treszezamsky et al., 2007*; *Figure 2—figure supplement 1E–F*). Similar results were observed using HCT116 cellular models depleted from BRCA1 (*Figure 2—figure supplement 2G–I*). The unique sensitivity of BRCA2-deficient cells to ROCKi suggests that the SL observed is likely independent of the homologous recombination function of BRCA2.

Importantly, we observed strong SL by ROCKi in growing conditions that triggered only mild sensitivity to PARPi. While HCT116$^{p21-/-}$ shBRCA2, V-C8 and DLD-1$^{BRCA2-/-}$ were all sensitive to olaparib (*Figure 2—figure supplement 3A*), PEO1 showed only modest sensitivity to olaparib in our experimental conditions (*Figure 2—figure supplement 3B*), despite reports indicating sensitivity to PARPi (*Sakai et al., 2009*; *Stukova et al., 2015*; *Whicker et al., 2016*). We confirmed that PEO1 were BRCA2-deficient. The BRCA2 mutation in PEO1 (5193C>G) creates a premature stop codon and also a digestion site for the enzyme DrdI. In contrast, the reversion mutation in PEO4 (5193C>T) abolishes this site (*Figure 2—figure supplement 3C*). Consistent with their expected point mutation, following DrdI digestion PEO1 cells showed two DNA fragments (480 bp and 214 bp), which were not observed in PEO4 cell lines (*Figure 2—figure supplement 3D*). Additionally, as previously reported for BRCA2-deficient cell lines (*Sakai et al., 2009*; *Stronach et al., 2011*; *Stukova et al., 2015*; *Whicker et al.,*

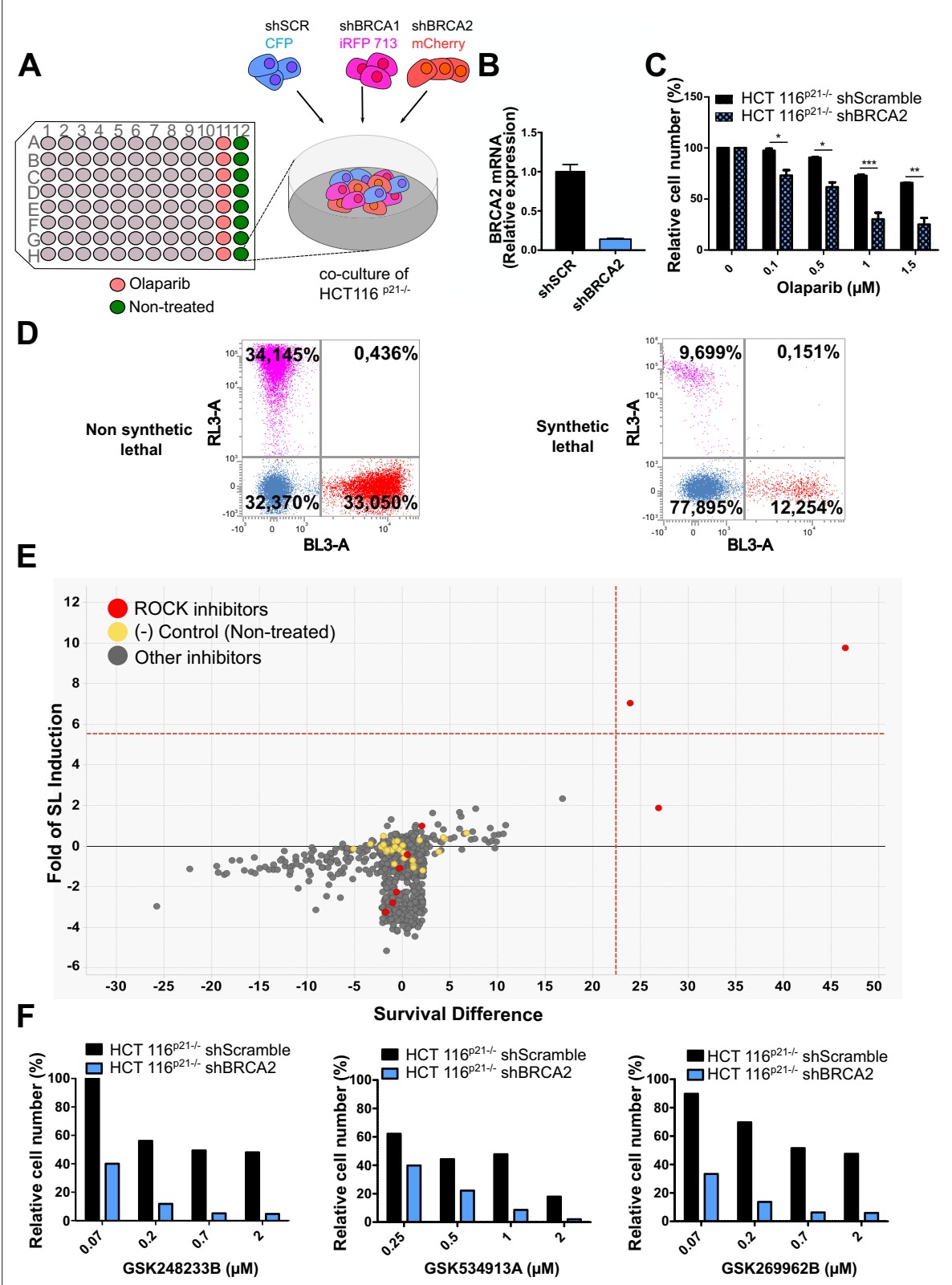

**Figure 1.** Phenotypic screening identifies ROCK kinases as potential targets for synthetic lethality in BRCA2 cells. (**A**) The screening assay is based on the co-culture of isogeneic BRCA-proficient and BRCA-deficient cell lines in equal proportions on each well of 96-well plates. Such cell lines were generated as double stable cell lines tagged with different fluorescent proteins (CFP, iRFP, and mCherry) and expressing shRNAs for Scramble, BRCA1 or BRCA2 were generated as described in **_Carbajosa et al., 2019_**. (**B**) Quantitative real-time PCR of BRCA2 in shScramble and shBRCA2 HCT116[p21-/-]

*Figure 1 continued*

cells (N=2). Statistical analysis was performed with a two-way ANOVA test followed by a Bonferroni post-test (*p<0.05, **p<0.01, ***p<0.001). (**C**) Relative cell number (%) of HCT116^p21-/- cells expressing shScramble and shBRCA2 and treated with the indicated concentrations of olaparib (N=2). (**D**) Representative results expressed as RL-3-BL-3 dot plots (log scale, RL-3 780/60 nm filter, and BL-3 695/40 nm filter). A tested compound can be 'non-synthetic lethal' (the ratio between the populations' percentage remains unchanged when compared to the ratio used for seeding ~33% for each cell line); or 'synthetic lethal' (the ratio between cell types is altered when compared to the ratio used for seeding, with selective depletion of cells within the BRCA1- and/or BRCA2-deficient populations). (**E**) Screening results of PKIS2 library compounds (0.1 μM) in shBRCA2 HCT116^p21-/- cells. Compounds were plotted based on their fold of SL (y axis) and their survival difference (x axis). A compound was considered a 'hit' if it exhibited a >5 standard deviations on these two variables. Fold of SL (y-axis): the ratios of the different populations in each individual well. Survival difference (x-axis): compares treated cells with the untreated control in the same plate. ROCK inhibitors and other inhibitors are plotted in red and gray, respectively. Please refer to *Carbajosa et al., 2019* for statistical analysis of the screening. (**F**) Relative cell number (%) of shScramble and shBRCA2 HCT116^p21-/- cells at different ROCK inhibitors. Data are shown as the average of independent experiments with the standard error of the mean.

The online version of this article includes the following source data for figure 1:

**Figure 1-source data 1.** Spreadsheet containing source data from *Figure 1*.

*2016*) PEO1 cells are sensitive to cisplatin (*Figure 2—figure supplement 3E*). Our results suggest that while clonogenic assays and other approaches may better expose the sensitivity of PEO1 to olaparib, strong SL induced by ROCKi is observed in growing conditions that reveal only mild sensitivity to PARPi. Hence, synthetic lethal avenues that diverge from PARPi could provide efficient therapeutic alternatives for treating BRCA2-deficient cancer cells.

## Replication stress is not the major driver of SL between BRCA2 deficiency and ROCK inhibition

The SL observed between BRCA deficiency and PARPi is preceded by the accumulation of acute replication stress caused by PARP trapping on the DNA (*Murai et al., 2012*; *Schoonen et al., 2017*). As BRCA-deficient cells keep progressing across S phase in the presence of PARPi, PARP/DNA adducts exacerbate replication stress resulting from fork stalling, gap formation and fork collapse (*Kolinjivadi et al., 2017*; *Lemaçon et al., 2017*; *Mijic et al., 2017*; *Panzarino et al., 2021*; *Schlacher et al., 2011*; *Taglialatela et al., 2017*). Consistent with those reports, the treatment of HCT116^p21-/- shBRCA2 cells with olaparib caused the acute accumulation of replication stress markers such as γH2AX and 53BP1 nuclear foci, which represent sites of DSB formation in S phase (*Figure 3A–C* and *Figure 3—figure supplement 1A–B*). In striking contrast to olaparib, no increase in 53BP1 or γH2AX foci was induced by fasudil treatment in HCT116^p21-/- shBRCA2 cells (*Figure 3A–C* and *Figure 3—figure supplement 1A–B*) at this time. These results were also validated in PEO cells (*Figure 3D* and *Figure 3—figure supplement 1C*). In line with the lack of acute replication stress, we did not observe alterations in DNA replication parameters, such as nascent DNA track length or the frequency of origin firing after fasudil treatment (*Figure 3F–G* and positive controls in *Figure 3—figure supplement 1D*). We also did not observe differences in the percentage of BrdU+ cells after 3 or 6 days of fasudil treatment compared to untreated cells (*Figure 3E*). Additionally, the intensity of BrdU, a parameter than reveals subtle alterations of the DNA replication program undetectable by the DNA fiber assay (*Calzetta et al., 2021*), was also unaffected (*Figure 3—figure supplement 1E*). Given that the synthetic lethality of fasudil was more evident 6 days post-treatment, we evaluated whether fasudil causes replicative stress at that time, and observed no evidence of augmented γH2AX intensity or 53BP1 focal organization in HCT116^p21-/- shBRCA2 and PEO1

**Table 1.** Phenotypic screening identifies ROCK kinases as potential targets for synthetic lethality in BRCA2 cells.

(A) Table listing all ROCK inhibitors from the PKIS2 library and their corresponding survival difference at 0.1 and 1 μM.

| Inhibitor | Survival difference | |
| --- | --- | --- |
| | 0.1 μM | 1 μM |
| GSK180736A | 0 | 8.15 |
| GSK248233B | 47.57 | 41.99 |
| GSK269962B | 25.58 | 28.49 |
| GSK270822A | 0 | 38.12 |
| GSK429286A | 0.29 | 18.11 |
| GSK466314A | 0 | 25.41 |
| GSK534911A | 25.5 | 33.72 |
| GSK534913A | 0 | 32.50 |
| SB-772077-B | 0 | 67.80 |

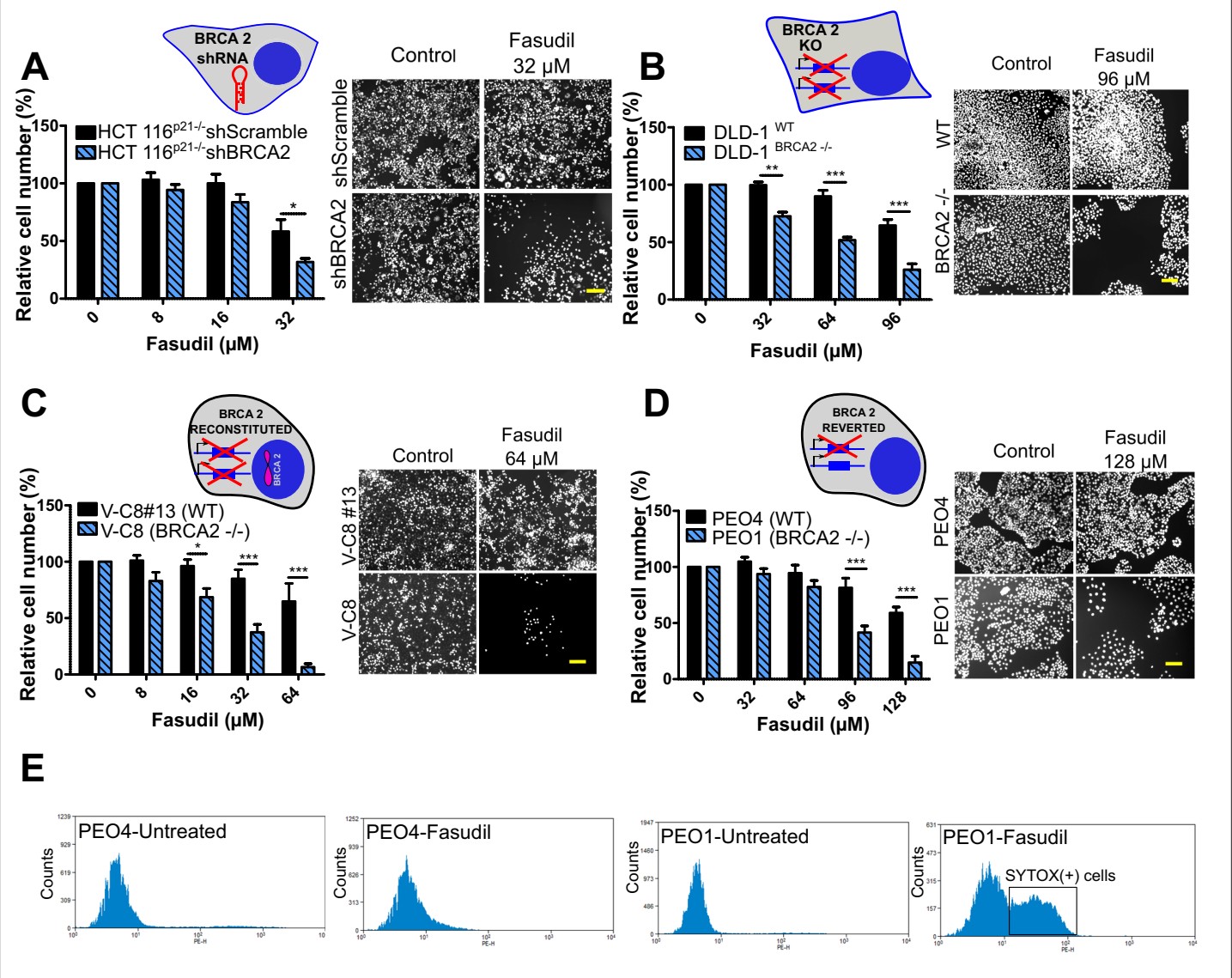

**Figure 2.** BRCA2-deficient cells are selectively killed by the ROCK kinase inhibitor fasudil. (**A**) Relative cell number (%) of shScramble and shBRCA2 HCT116[p21-/-] cells after 6 days of treatment with fasudil (N=3). (**B**) Relative cell number (%) of DLD-1[WT] and DLD-1[BRCA2-/-] after 6 days of treatment with fasudil (N=2). (**C**) Relative cell number (%) of V-C8#13 and V-C8 cells after 6 days of treatment with fasudil (N=3). (**D**) Relative cell number (%) of PEO4 and PEO1 cells after 6 days of treatment with fasudil (N=4). Panels A-D: the cell cartoon shows the BRCA2 status caused by the modification introduced at last in each pair of cell lines (see Materials and methods for further details). Black borders indicate that the modification generated a BRCA2 proficient status and blue borders aBRCA2 deficiency. (**E**) FACS analysis of SYTOX green-stained PEO4 and PEO1 cells 6 days after fasudil treatment (128 µM, N=2). Statistical analysis was performed with a two-way ANOVA test followed by a Bonferroni post-test (*p<0.05, **p<0.01, ***p<0.001). Data in A-D are shown as the average of independent experiments with the standard error of the mean.

The online version of this article includes the following source data and figure supplement(s) for figure 2:

**Figure 2-source data 1.** Spreadsheet containing source data from *Figure 2*.

**Figure supplement 1.** BRCA2-deficient cells are sensitive to the ROCK kinase inhibitor, fasudil.

**Figure 2-figure supplement 1-source data 1.** Spreadsheet containing source data from *Figure 2—figure supplement 1*.

**Figure supplement 2.** BRCA2-deficient cells are sensitive to the ROCK kinase inhibitor ripasudil.

**Figure 2-figure supplement 2-source data 1.** Spreadsheet containing source data from *Figure 2—figure supplement 2*.

**Figure supplement 3.** BRCA2-deficient cells are sensitive to olaparib.

**Figure 2-figure supplement 3-source data 1.** Spreadsheet containing source data from *Figure 2—figure supplement 3* (source data 1).

**Figure 2-figure supplement 3-source data 2.** Spreadsheet containing source data from *Figure 2—figure supplement 3* (source data 2).

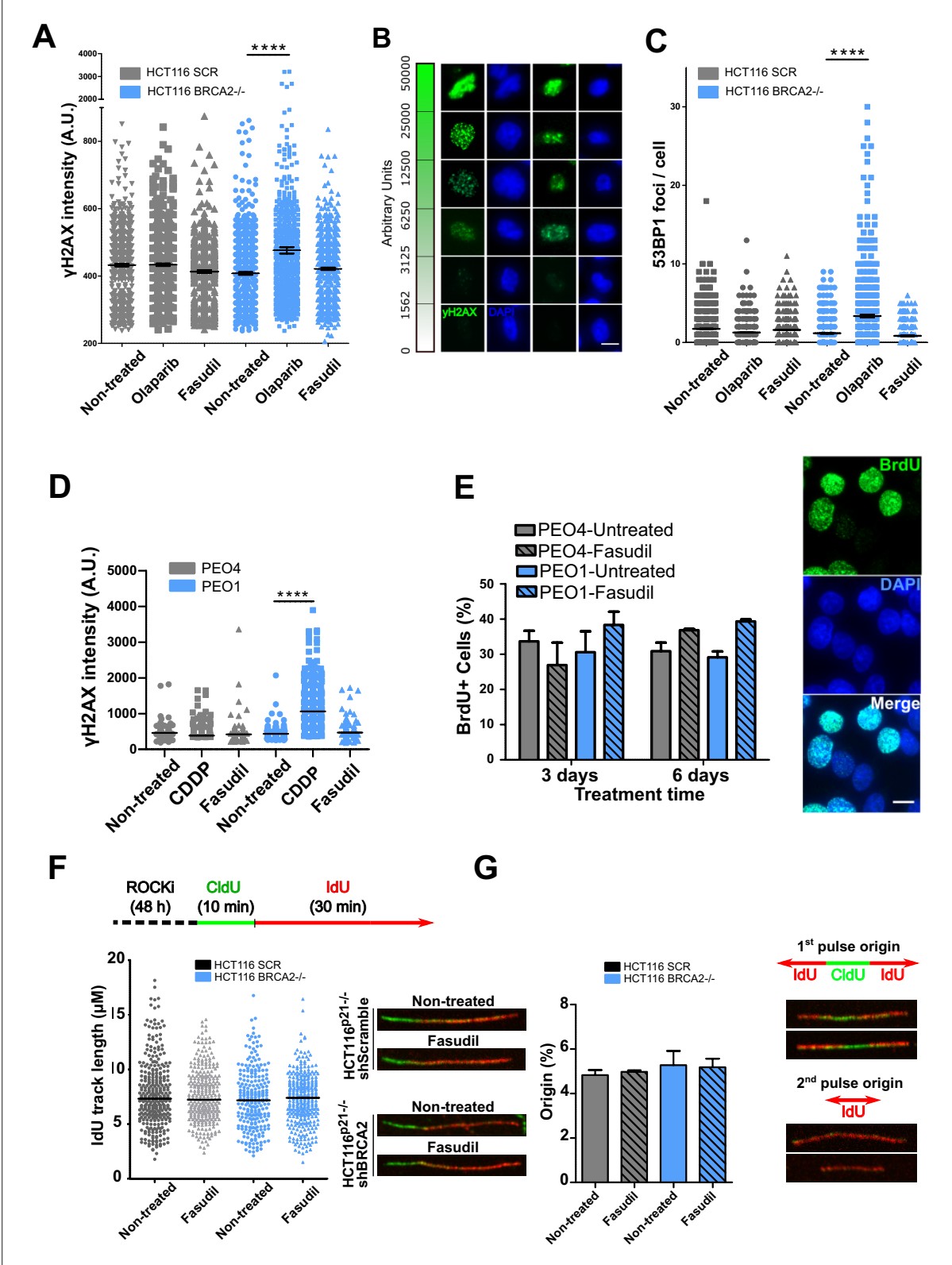

**Figure 3.** Fasudil does not induce acute replication stress in BRCA2-deficient cells. (**A**) yH2AX intensity/cell of shScramble or shBRCA2 HCT116[p21-/-] cells (N=2). (**B**) Representative images of yH2AX intensity in single cells. (**C**) Number of 53BP1 foci/cell in shScramble and shBRCA2 HCT116[p21-/-] cells (N=2). (**D**) yH2AX intensity/cell in PEO1 or PEO4 cells (N=2). (**E**) Percentage of PEO4 and PEO1 cells stained with BrdU at 3 and 6 days after fasudil treatment (128 μM, N=2). A total of 500 cells were analyzed for each sample. Representative images of PEO1 cells after 3 days of fasudil treatment (BrdU shown

*Figure 3 continued on next page*

*Figure 3 continued*

in green, DAPI shown in blue). (**F**) Labelling scheme and IdU track lengths of shScramble and shBRCA2 HCT116$^{p21-/-}$ cells, treated with fasudil for 48 h (N=2). Representative images of individual DNA fibers are shown on the left side of the panel. (**G**) Origin firing frequency (percentage) of shScramble or shBRCA2 HCT116$^{p21-/-}$ cells in samples showed in E (N=2). Statistical analysis was performed using a two-way ANOVA test followed by a Bonferroni post-test (*p<0.05, **p<0.01, ***p<0.001). Data are shown as the average of independent experiments with the standard error of the mean.

The online version of this article includes the following source data and figure supplement(s) for figure 3:

**Figure 3-source data 1.** Spreadsheet containing source data from *Figure 3* (source data 1).

**Figure 3-source data 2.** Spreadsheet containing source data from *Figure 3* (source data 2).

**Figure 3-source data 3.** Spreadsheet containing source data from *Figure 3* (source data 3).

**Figure 3-source data 4.** Spreadsheet containing source data from *Figure 3* (source data 4).

**Figure supplement 1.** Fasudil does not alter S phase parameters in BRCA2-deficient cells.

**Figure 3-figure supplement 1-source data 1.** Spreadsheet containing source data from *Figure 3—figure supplement 1* (source data 1).

**Figure 3-figure supplement 1-source data 2.** Spreadsheet containing source data from *Figure 3—figure supplement 1* (source data 2).

**Figure 3-figure supplement 1-source data 3.** Spreadsheet containing source data from *Figure 3—figure supplement 1* (source data 3).

**Figure 3-figure supplement 1-source data 4.** Spreadsheet containing source data from *Figure 3—figure supplement 1* (source data 4).

**Figure supplement 2.** Fasudil does not induce acute replication stress after 6 days of treatment in BRCA2-deficient cells.

**Figure 3-figure supplement 2-source data 1.** Spreadsheet containing source data from *Figure 3—figure supplement 2*.

**Figure 3-figure supplement 2-source data 2.** Spreadsheet containing source data from *Figure 3—figure supplement 2* (source data 2).

**Figure 3-figure supplement 2-source data 3.** Spreadsheet containing source data from *Figure 3—figure supplement 2* (source data 3).

**Figure 3-figure supplement 2-source data 4.** Spreadsheet containing source data from *Figure 3—figure supplement 2* (source data 4).

(BRCA2-/-) 6 days post-treatment (*Figure 3—figure supplement 2*). These findings point toward a cell death mechanism independent from the accumulation of DNA damage in S phase.

## ROCK inhibition induces mitotic defects in BRCA2-deficient cells

To further characterize such a replication stress-independent SL, we analyzed cell cycle profiles with propidium iodide staining. Consistent with reduced survival at 6 days (*Figure 2*), in BRCA2-deficient cells, we observed a sub-G1 peak after fasudil treatment indicative of apoptotic cell death (*Figure 4A–B*). In terms of cell cycle distribution, BRCA2-deficient cells treated with fasudil exhibited an accumulation of cells in G2/M indicative of a G2/M arrest (*Figure 4A–B*). Intriguingly, BRCA2-deficient cells also exhibited a peak of >4N polyploid cells (*Figure 4A–B*). By performing a detailed time course in which samples were collected in 24 h intervals, we observed that the polyploidy phenotype was cumulative (*Figure 4C*). While the G2/M arrest in BRCA2-deficient cells appeared as early as 24 h post-treatment, polyploidy became strongly evident after 72 h (i.e.: 3 days). The sub-G1 population was also evident as early as 24 h post-treatment but increased at longer time points after polyploidy detection (i.e.: after 3 days). These data suggest that the accumulation of cells in G2/M precedes both polyploidy and cell death.

The concomitant accumulation of cells in G2/M (which could also include G1 cells with duplicated DNA content) and the DNA content >4N is highly suggestive of problems in the correct finalization of M phase, which leads to the accumulation of aberrant mitotic phenotypes. Consistent with this, after fasudil treatment, BRCA2-deficient cells exhibited an increase in metaphases in which the DNA was being pulled in multiple directions or in which the chromosomes were not aligned in the metaphase plate (*Figure 4D–E*). Altogether, these data pinpoint a dysregulated mitosis in BRCA2-deficient cells treated with ROCKi.

Aberrant metaphases can be triggered by unresolved DNA replication defects accumulated after DNA replication stress (*Gelot et al., 2015*), but can also be prompted within M phase as a consequence of aberrant mitotic spindle organization or disorganized chromosome alignment (*Bakhoum et al., 2009*; *Shindo et al., 2021*; *Siri et al., 2021*). Aberrant anaphases (bridges and lagging chromosomes; *Figure 5A*) can also be triggered either by replication defects not resolved before M phase entry or intrinsic mitotic defects dissociated from S phase (*Bakhoum et al., 2009*; *Shindo et al., 2021*). We documented an increase in chromosome bridges, but not in lagging chromosomes, after fasudil treatment of BRCA2-deficient cells (*Figure 5B–C*). To confirm the increase of chromosome

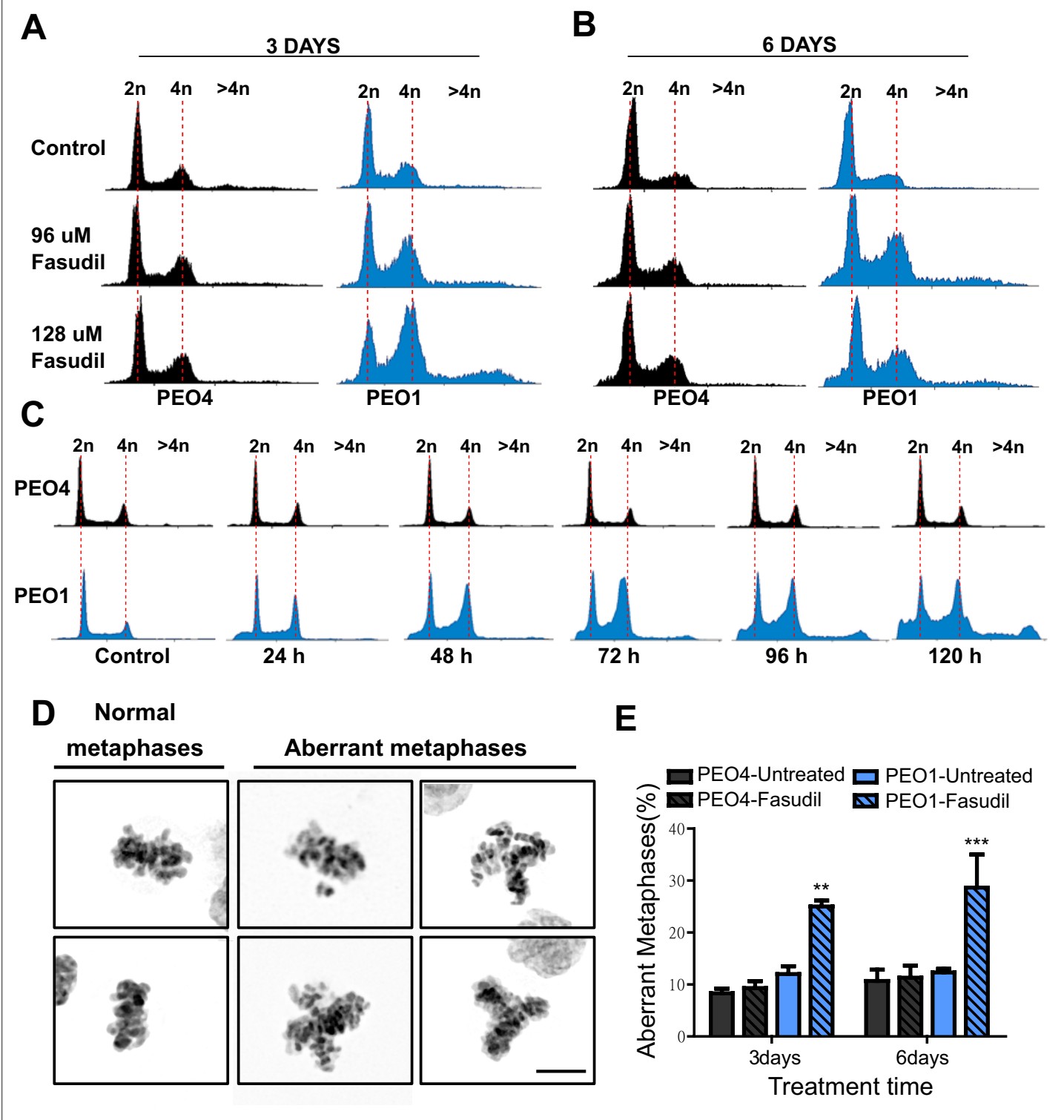

**Figure 4.** Fasudil treatment induces polyploidy and aberrant mitotic figures in BRCA2-deficient cells. (**A–B**) Cell cycle analysis of PEO4 and PEO1 cells following 3 or 6 days of fasudil treatment (96 and 128 μM; N=3). Cells were stained with propidium iodide, and DNA content was analyzed via FACS (10,000 events per sample). (**C**) Cell cycle analysis of PEO4 and PEO1 cells following a time course with fasudil treatment (N=2; 1–5 days, 64 μM). Cells were stained with propidium iodide, and DNA content was analyzed via FACS (10,000 events per sample). (**D**) Representative images of DAPI-stained normal and aberrant metaphases. Aberrant metaphases include metaphases with DNA being pulled in multiple directions or metaphases with misaligned chromosomes. (**E**) Percent of aberrant metaphases in PEO4 and PEO1 cells 3 or 6 days after fasudil treatment (128 μM; N=3). A total of 100

*Figure 4 continued on next page*

*Figure 4 continued*

metaphases were analyzed for each sample. Statistical analysis was performed using a two-way ANOVA test followed by a Bonferroni post-test (*$p<0.05$, **$p<0.01$, ***$p<0.001$). Data are shown as the average of independent experiments with the standard error of the mean.

The online version of this article includes the following source data for figure 4:

**Figure 4-source data 1.** Spreadsheet containing source data from *Figure 4*.

bridges observed with fasudil, we used commercially available siRNAs against ROCK1 and ROCK2 (*Figure 5D*). Similarly to ROCKi, ROCK1 and ROCK2 (ROCK1/2) depletion promoted the accumulation of anaphase bridges in BRCA2-deficient cells (*Figure 5E*). Importantly, when resulting from unresolved replication defects, anaphase aberrations are typically accompanied by chromosome aberrations (i.e. breaks, exchanges) and micronuclei (*Finardi et al., 2020*; *Utani et al., 2010*). However, we did not find any indication of chromosome aberrations or micronuclei in fasudil-treated BRCA2-deficient cells (*Figure 5—figure supplement 1A–B*), suggesting that the trigger for anaphase bridge formation following fasudil treatment is a defect intrinsic to M phase.

## ROCK inhibition causes cytokinesis failure in BRCA2-deficient cells

Since BRCA2-deficient cells treated with ROCKi accumulate M phase defects, we explored the link between ROCK and mitosis. ROCK are crucial regulators of the actin cytoskeleton (*Julian and Olson, 2014*) and play a role in cleavage furrow formation during cytokinesis (*Kosako et al., 2000*; *Yokoyama et al., 2005*). BRCA2 was also implicated in regulating the contraction of the actin cytoskeleton towards the end of mitosis and its downregulation or absence induces multinucleation due to cytokinesis failure (*Daniels et al., 2004*; *Jonsdottir et al., 2009*; *Mondal et al., 2012*; *Shive et al., 2010*; *Vinciguerra et al., 2010*). Moreover, BRCA2 localizes to the midbody during cytokinesis (*Daniels et al., 2004*; *Jonsdottir et al., 2009*; *Mondal et al., 2012*; *Rowley et al., 2011*; *Takaoka et al., 2014*) and its downregulation or absence was also reported to induce multinucleation (*Lekomtsev et al., 2010*). To explore whether a convergent defect triggers cytokinesis failure after ROCK inhibition in BRCA2-deficient cells, we stained the actin cytoskeleton with phalloidin to distinguish the cytoplasm of individual cells and analyzed the formation of binucleated as well as multinucleated cells after fasudil treatment (*Figure 6A*). We observed a marked increase of binucleation in BRCA2-deficient cells following fasudil treatment (*Figure 6B–C*). Also, we documented an increase of multinucleation in BRCA2-deficient cells transfected with siROCK (*Figure 6—figure supplement 1A–B*). Consistent with the polyploidy (>4N) observed with flow cytometry, fasudil treatment also increased the percentage of multinucleated cells with 3, 4, or 5+ nuclei (*Figure 6B–C*). Similar to the polyploidy observed in the cell cycle profiles, the proportion of multinucleated cells was more severe at later endpoints (*Figure 6B–C*), suggesting that despite cytokinesis failure, binucleated cells continue to cycle, thus further increasing their DNA content. Indeed, the percentage of BRCA2-deficient binucleated cells transiting S phase, as revealed by cyclin A staining, was between 30 and 40% irrespective of ROCKi. This result indicates that despite their diploid DNA content, BRCA2-deficient cells treated with fasudil were able to start a new cell cycle and transit through a second S phase (*Figure 6—figure supplement 1C–D*).

One immediate consequence of cytokinesis failure is that the resulting cell contains two centrosomes instead of one (*Ganem et al., 2007*). Normal cells harbor one centrosome, which duplicates only once during S phase. During normal mitosis, duplicated centrosomes form a bipolar mitotic spindle ensuring equal chromosome distribution in daughter cells (*Nigg, 2007*). In contrast, multiple centrosomes can lead to multipolar mitosis and cell death (*Ganem et al., 2009*). We stained cells for gamma-tubulin and alpha-tubulin, central components of centrosomes and microtubules, respectively (*Brinkley, 1997*; *Fuller et al., 1995*) and focused the analysis on mitotic cells. BRCA2-deficient cells treated with fasudil exhibited increased numbers of multipolar mitosis that correlated with increased centrosome number (i.e.:>2; *Figure 6D–F*). Similar to previously observed phenotypes, such as aberrant metaphases, binucleated cells and polyploidy, the percentage of multipolar mitosis increased at later endpoints (*Figure 6F*). Together, these results suggest that the cytokinesis failure and altered centrosome numbers lead to multipolar mitosis, which could trigger cell death in fasudil-treated BRCA2-deficient cells.

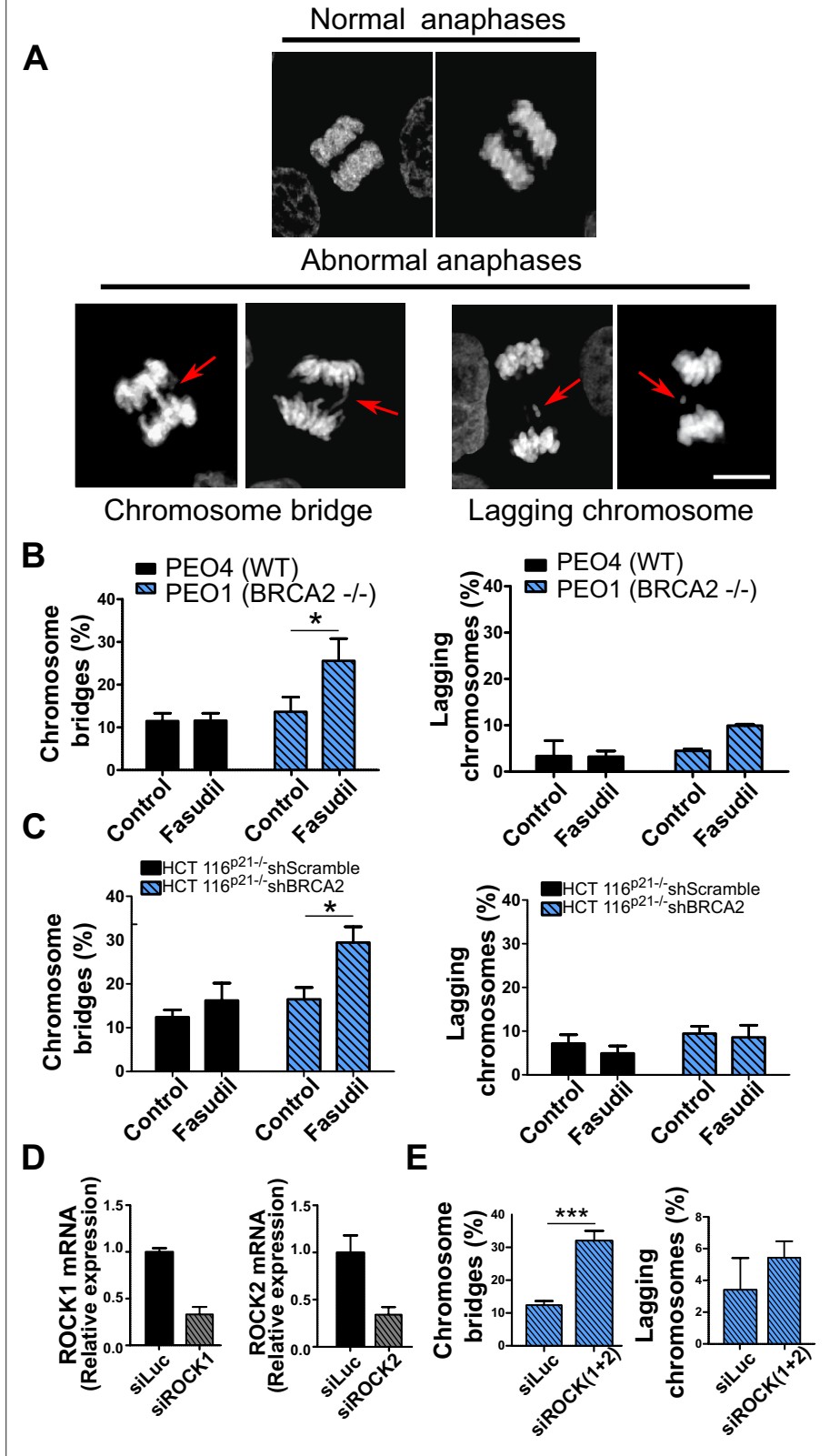

**Figure 5.** Mitotic DNA bridges accumulate in BRCA2-deficient cells following ROCK inhibition with fasudil. (**A**) Representative images of normal and abnormal anaphases with bridges and lagging chromosomes. (**B**) Percentage of anaphases with chromosomes bridges and lagging chromosomes in PEO4 and PEO1 cells treated with fasudil (128 µM). Fifty to 70 anaphases per sample were analyzed in two independent experiments (N=2). (**C**) Percentage

*Figure 5 continued on next page*

*Figure 5 continued*

of anaphases with chromosomes bridges and lagging chromosomes in shScramble- or shBRCA2-transduced HCT116$^{p21-/-}$ cells treated with fasudil. 50–70 anaphases per sample were analyzed per independent experiment (N=3). (**D**) Quantitative real-time PCR of ROCK1 and ROCK2 in shBRCA2 HCT116$^{p21-/-}$ cells transfected with 150 µM of siROCK1 or siROCK2 (N=2). (**E**) Percentage of anaphases with chromosomes bridges and laggards in shBRCA2 HCT116$^{p21-/-}$ cells transfected with siROCK (1+2). A total of 50–70 anaphases per sample were analyzed in three independent experiments (N=2). The statistical analysis of the data was performed with a two-way ANOVA test followed by a Bonferroni post-test (*$p<0.05$, **$p<0.01$, ***$p<0.001$). Data are shown as the average of independent experiments with the standard error of the mean.

The online version of this article includes the following source data and figure supplement(s) for figure 5:

**Figure 5-source data 1.** Spreadsheet containing source data from *Figure 5*.

**Figure supplement 1.** BRCA2-deficient cells treated with fasudil do not display replication stress-derived chromosome defects.

**Figure 5-figure supplement 1-source data 1.** Spreadsheet containing source data from *Figure 5—figure supplement 1*.

## Cytokinesis failure sensitizes BRCA2-deficient cells to cell death

The results described in *Figure 4A–C*, *Figure 6—figure supplement 1C–D*, and *Figure 6* indicate that the treatment of BRCA2-deficient cells with ROCKi causes cytokinesis failure and triggers the accumulation of binucleated cells with proliferation capacity. The implications are that cells with >4N DNA content die when attempting to duplicate aberrantly duplicated DNA or when assembling aberrant mitotic spindles in the subsequent mitosis. Supporting such a model is the time course in *Figure 7A–C*. A significant change in the binucleation of BRCA2-deficient cells was observed as early as 24 h post-fasudil (*Figure 7A*), while a significant increase of aberrant anaphases and mitosis was detected later on, at 48 h (*Figure 7B–C*). Surprisingly, binucleation-related cell death is not triggered in control cells, even at doses of ROCKi that kill BRCA2-proficient cells (*Figure 7—figure supplement 1A–B*). Hence, these results support the likelihood of cytokinesis failure as the trigger for the SL caused by ROCK inhibition in BRCA2-deficient cells.

If cytokinesis defects caused by ROCKi are the trigger of BRCA2-deficient SL, targeting other factors that regulate cytokinesis should also induce cell death. To test this hypothesis, we downregulated Citron Rho-interacting kinase (CITK), an enzyme that is highly enriched in the midbody during cytokinesis (*Madaule et al., 1998*; *Sahin et al., 2019*; *Figure 8A*). CITK is required for proper RhoA localization at the cleavage site during late cytokinesis (*Sahin et al., 2019*). Similar to the phenotypes of siROCK1/2, CITK downregulation reduced cell survival of BRCA2-deficient cells (*Figure 8B* and *Figure 8—figure supplement 1A*). In addition, and recapitulating the effect of ROCK inhibition or depletion, CITK downregulation increased the number of multinucleated cells in BRCA2-deficient cells (*Figure 8C*). Most remarkably, combined silencing of CITK and ROCK1/2 was not additive/synergic (*Figure 8B*), suggesting that ROCK and CITK depletion induce synthetic lethality in BRCA2-deficient cells. Together, these findings indicate that cytokinesis failure by multiple sources could induce death in BRCA2-deficient cells.

If aberrant transit through mitosis is the origin of the cell death triggered by ROCKi, then the bypass of mitosis should protect those cells from cell death. To this end, we downregulated Early mitotic inhibitor-1 (EMI1), an anaphase-promoting complex (APC) inhibitor that has a crucial role in the accumulation of mitosis activators, including B-type cyclins (*Reimann et al., 2001*). When transfecting siEMI1, we observed a 65% reduction in EMI1 expression (*Figure 7D*) and, as reported by others (*Robu et al., 2012*; *Shimizu et al., 2013*; *Verschuren et al., 2007*), accumulation of cells with G2/M DNA content or higher (*Figure 8E*). EMI1 depletion prevented the SL effect of ROCKi on different BRCA2-deficient cells (*Figure 8F* and *Figure 8—figure supplement 1B*). Therefore, these results indicate that BRCA2-deficient cells that die upon ROCK inhibition do so after transiting an aberrant mitosis.

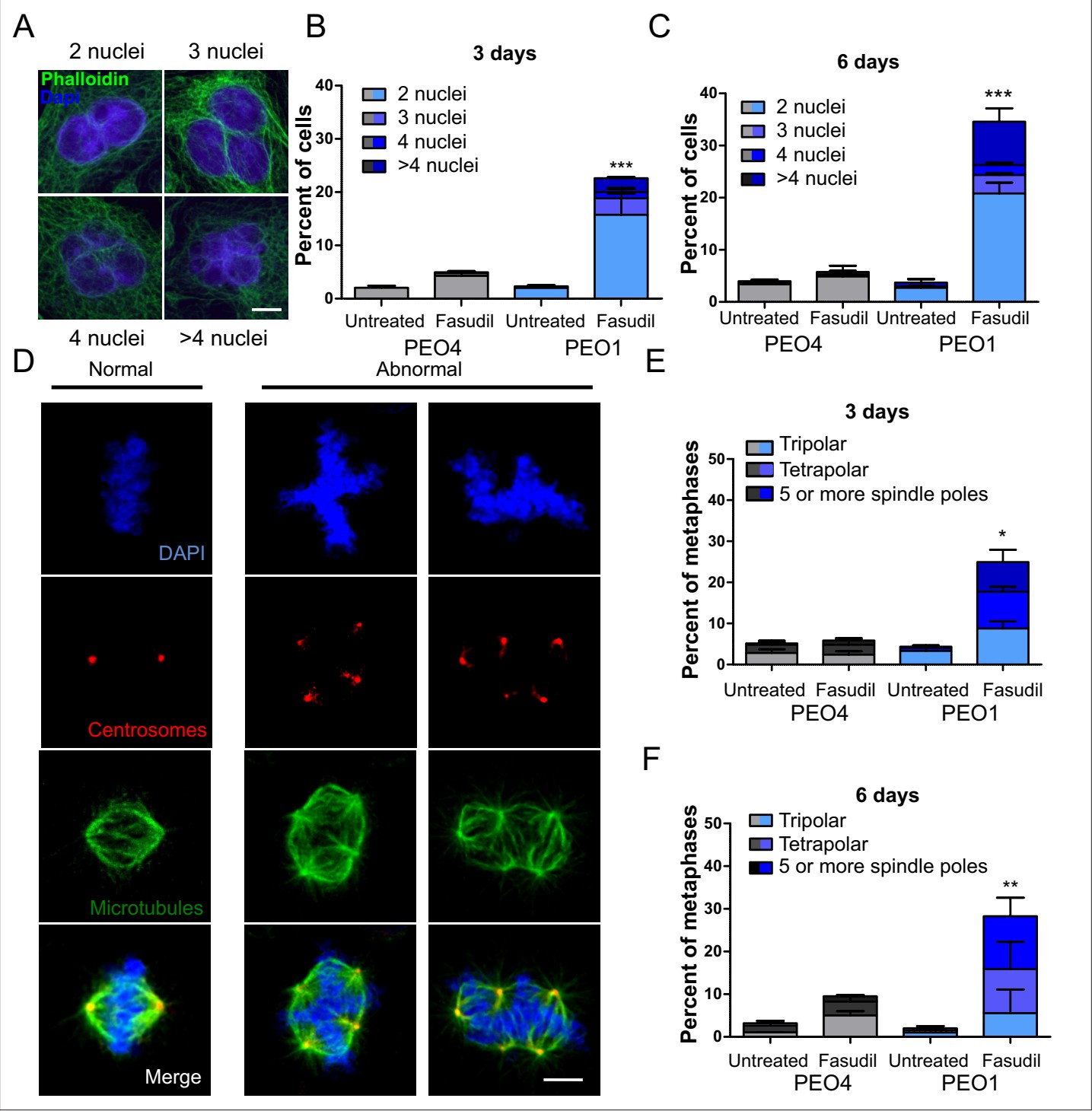

**Figure 6.** BRCA2-deficient cells exhibit cytokinesis failure, centrosome amplification and multipolar mitotic spindles following fasudil treatment. (**A**) Representative pictures of PEO1 cells after fasudil treatment. Nuclei are stained with DAPI (shown in blue), and the cytoplasm of individual cells is stained with phalloidin which stains the actin cytoskeleton (shown in green). (**B**) Percent of binucleated and multinucleated PEO4 and PEO1 cells after 3 days of fasudil treatment (N=3, 128 µM). (**C**) Percent of binucleated and multinucleated number of PEO4 and PEO1 cells after 6 days of fasudil treatment (N=3, 128 µM). A total of 200 cells were analyzed per sample. (**D**) Representative pictures of PEO1 metaphases showing cells with normal and abnormal mitotic spindles. DNA, centrosomes, and microtubules are shown in blue, red, and green, respectively. (**E**) Percent of metaphases in PEO4 and PEO1 cells with multipolar spindles after 3 days of fasudil treatment (N=3, 128 µM). (**F**) Percent of metaphase in PEO4 and PEO1 cells with multipolar spindles after 6 days of fasudil treatment (N=2, 128 µM). Mitotic spindles were visualized by staining centrosomes (γ-tubulin) and microtubules (α-tubulin) and DNA was stained with DAPI. Cells were classified as having multipolar spindle (3, 4, or 5 or more spindles). A total of 100 metaphases

*Figure 6 continued on next page*

*Figure 6 continued*

were analyzed per sample. Statistical analysis was performed using a two-way ANOVA test followed by a Bonferroni post-test (*p<0.05, **p<0.01, ***p<0.001). Data are shown as the average of independent experiments with the standard error of the mean.

The online version of this article includes the following source data and figure supplement(s) for figure 6:

**Figure 6-source data 1.** Spreadsheet containing source data from *Figure 6*.

**Figure supplement 1.** Multinucleated BRCA2-deficient cells resulting from fasudil treatment are able to transit through S phase.

**Figure 6-figure supplement 1-source data 1.** Spreadsheet containing source data from *Figure 6—figure supplement 1*.

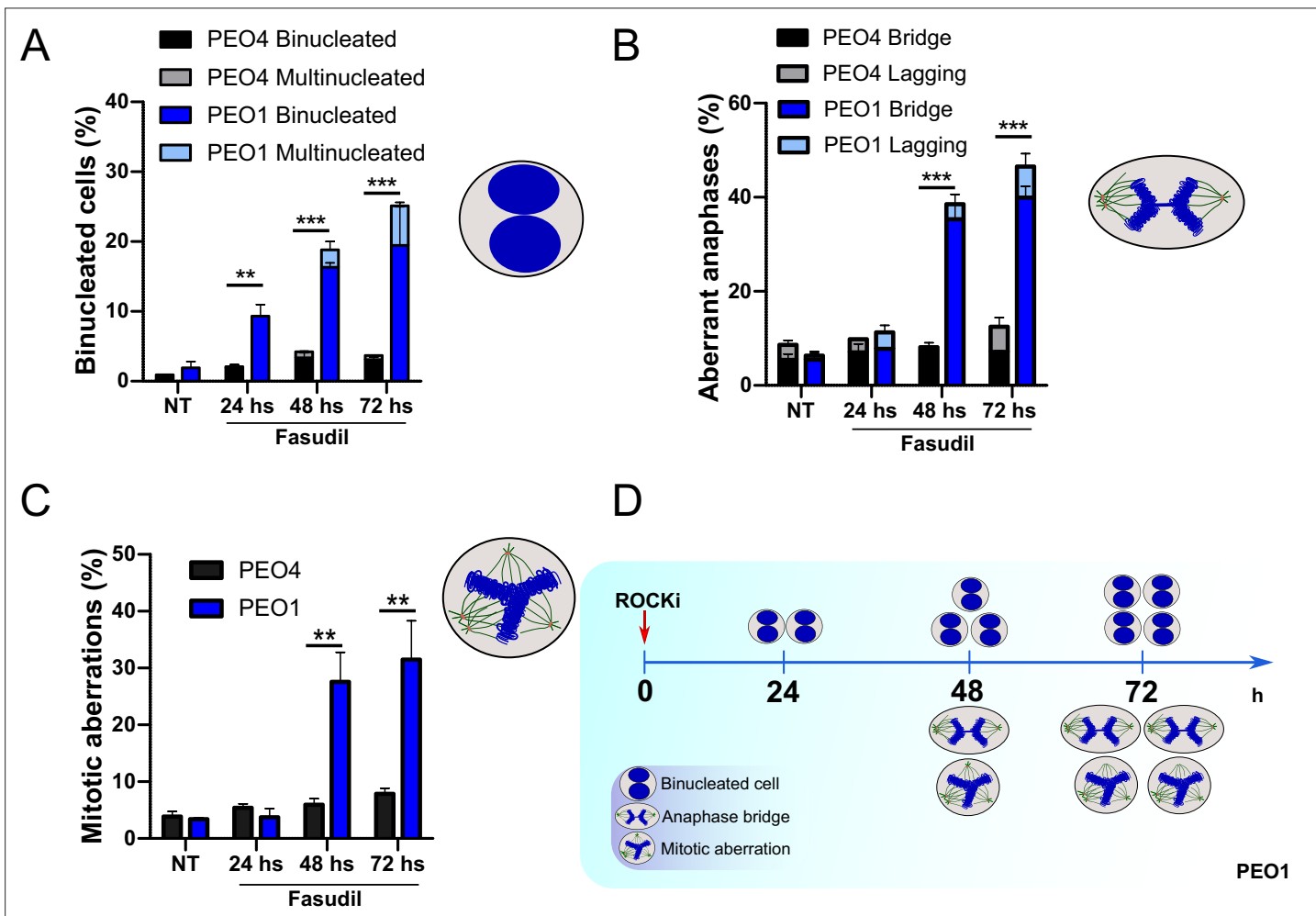

**Figure 7.** Binucleation precedes anaphase and mitotic aberrations in BRCA2-deficient cells. (**A**) Percent of binucleated PEO1 and PEO4 cells treated with fasudil at the indicated time points after treatment (N=2). (**B**) Percent of aberrant anaphases in PEO1 and PEO4 cells treated with fasudil at the indicated time points after treatment (N=2). (**C**) Percent of mitotic aberrations in PEO1 and PEO4 cells treated with fasudil at the indicated time points after treatment (N=2). For panels A to C, statistical analysis was performed using a two-way ANOVA test followed by a Bonferroni post-test (*p<0.05, **p<0.01, ***p<0.001). Data are shown as the average of independent experiments with the standard error of the mean. (**D**) Representative scheme of the results obtained in **A-C**.

The online version of this article includes the following source data and figure supplement(s) for figure 7:

**Figure 7-source data 1.** Spreadsheet containing source data from *Figure 7*.

**Figure supplement 1.** Binucleation-related cell death is not triggered in BRCA2-proficient cells.

**Figure 7-figure supplement 1-source data 1.** Spreadsheet containing source data from *Figure 7—figure supplement 1*.

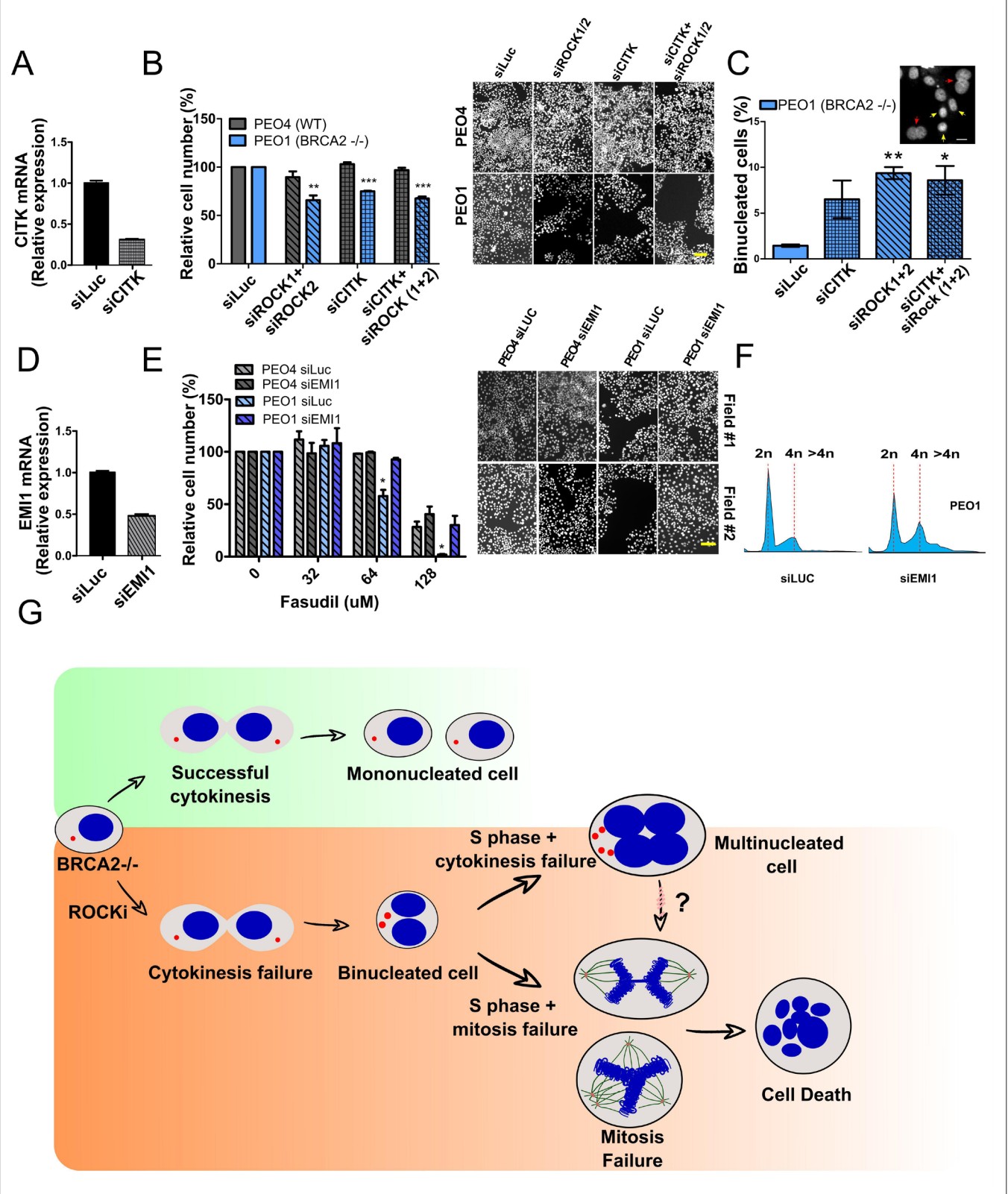

**Figure 8.** Mitosis as an alternative synthetic lethality strategy for BRCA2-deficient cells. (**A**) Quantitative real-time PCR of CITK in shBRCA2 HCT116[p21-/-] cells transfected with 150 µM of siCITK (N=2). (**B**) Relative cell number (%) of PEO4 and PEO1 after 6 days of being transfected with siROCK (1+2), siCITK or siROCK (1+2)/siCITK and representative images of the transfected cells (N=2). (**C**) Percent of binucleated PEO1 cells transfected with siROCK (1+2), CITK or siROCK (1+2)/siCITK (N=2). (**D**) Quantitative real-time PCR of EMI1 in shBRCA2 HCT116[p21-/-] cells transfected with 150 µM of siEMI1 (N=2). (**E**)

*Figure 8 continued on next page*

*Figure 8 continued*

Relative cell number (%) of PEO4 and PEO1 after 6 days of being transfected with siEMI1 and treated with fasudil (N=2). Representative images of the transfected and treated cells. (**F**) Cell cycle analysis of PEO1 cells following transfection with siEMI1 for 48hs (N=2). Cells were stained with propidium iodide and DNA content was analyzed via FACS (10,000 events per sample). (**G**) Model depicting the events leading to BRCA2-deficient cell death after fasudil treatment. The inhibition or depletion of ROCK in BRCA2-deficient cells leads to cytokinesis failure. As a result, the daughter cells are binucleated (4N) and have extra centrosomes (two instead of one). We speculate that after a subsequent DNA duplication, these cells can attempt mitosis. Mitosis entry with increased DNA content and extra centrosomes may frequently give rise to abnormal and multipolar spindles, leading to misaligned chromosomes and mitotic failure due to multipolar spindle formation. Alternatively, cytokinesis may fail again, and cells may temporarily survive as multinucleated cells, possibly facing cell death during subsequent mitotic attempts.

The online version of this article includes the following source data and figure supplement(s) for figure 8:

**Figure 8-source data 1.** Spreadsheet containing source data from *Figure 8*.

**Figure supplement 1.** Mitosis as an alternative synthetic lethality strategy for BRCA2-deficient cells.

**Figure 8-figure supplement 1-source data 1.** Spreadsheet containing source data from *Figure 8—figure supplement 1*.

## Discussion

### Targeting mitosis as an alternative SL strategy

In this work, we used a novel screening platform developed and validated by our group (*Carbajosa et al., 2019*; *García et al., 2020*) to identify ROCK as novel targets for SL induction in BRCA2-deficient cells. Using commercially available, and clinically relevant, ROCKi (i.e.: fasudil, ripasudil and SR 3677 dihydrochloride) (*Feng et al., 2016*; *Feng et al., 2008*; *Lee et al., 2019*), we observed a dose-dependent SL-induction in multiple BRCA2-deficient cell lines which showed no signs of DNA replication stress. In contrast, these cells exhibited strong mitotic defects due to the cytokinesis failure induced by ROCKi. Remarkably, cell death by ROCK inhibition or depletion was recapitulated by inhibiting another enzyme that facilitates cytokinesis, CITK, supporting a model in which binucleation precedes multinucleation and SL (*Figure 8G*). In fact, robust evidence in the literature indicates that highly abnormal metaphases/anaphases, such as the ones we observed, are incompatible with cell viability (*Ganem et al., 2009*) and are, therefore, the most plausible cause for the SL induced by ROCKi in BRCA2-deficient cells. While still viable, multinucleated cells are highly vulnerable. The presence of extra DNA content and centrosomes, increases the chances of abnormal spindle polarity, as well as the number of chromosomes that need to be properly aligned. In fact, attempts to trigger cell division in such states is incompatible with viability (*Ganem et al., 2009*; *Dale Rein et al., 2015*; *Schoonen et al., 2017*). We, therefore, postulate that the cytokinesis failure of a cell with 4N or more DNA content is the major driver for BRCA2-deficient cell death following ROCK inhibition. Because we have not identified the molecular target of ROCK which dysregulation triggers SL in BRCA2-deficient cells, further research on the mitotic functions of BRCA2 will be necessary to fully understand this SL pathway. However, we believe it is valuable to report that targeting mitosis alone in the absence of increased replication stress may suffice to kill BRCA2-deficient cells.

### BRCA2-deficient cells can be killed in a manner that is independent from the induction of replication stress

In addition to the well-documented replication stress-mediated toxicity of PARPi in BRCA-deficient cells, a recent report indicates that BRCA2-deficient cells can also be killed by mild replication defects which do not cause γH2AX accumulation in S phase (*Adam et al., 2021*). This cell death depends on the transmission of under-replicated DNA from S to M phase triggered by BRCA1 or BRCA2 deficiency and the lack of CIP2A-TOPBP1 complex formation in M phase. In the absence of this complex, under-replicated DNA is aberrantly processed into acentric chromosomes and micronuclei, which are the source of SL (*Adam et al., 2021*). Our present work reveals yet another weakness of BRCA2-deficient cells: cytokinesis. Strikingly, this SL is not preceded by the accumulation of broken chromosomes or micronuclei and is independent of canonical players of the DDR, as it is observed after ROCK or CITK inhibition.

Intriguingly, while the triggers of SL by PARPi, CIP2A-TOPBP1 complex disruption and ROCKi are remarkably different, the three mechanisms converge at mitosis; see *Adam et al., 2021*; *Schoonen et al., 2017*; *Schoonen and van Vugt, 2018*; and this work. CDK1 inhibition blocks micronucleation which is the trigger for BRCA-deficient cell death by CIP2A-TOPBP1 complex disruption (*Adam et al.,*

*2021*), while PARPi and ROCKi-mediated cell death is abrogated by EMI1-depletion- see *Schoonen et al., 2017* and this work. Hence, the transit through M phase is required for all SL events triggered in BRCA2-deficient cells. Of note, the accumulation of viable multinucleated BRCA2-depleted cells capable of enabling DNA synthesis after ROCKi reveal that, at least for a few DNA replication cycles, a cytokinesis-free cell cycle progression does not affect survival of BRCA2-deficient cells. Interestingly, multinucleation was also reported after PARPi treatment (*Schoonen et al., 2017*) and anaphase bridges were detected both after ROCKi and PARPi as a potential source of either multinucleation or cell death- see *Schoonen et al., 2017* and this work. In conclusion, despite the difference in the initial trigger of cell death, both after PARPi and ROCKi, BRCA2-deficient cells die at mitosis.

It should also be mentioned that our experimental analysis does not rule out that background levels of replication stress or increased levels of under-replicated DNA induced by BRCA2 deficiency could contribute to the cell death triggered by ROCK inhibition. As previously suggested by *Adam et al., 2021*, it is possible that BRCA2-deficient cells rely more on M phase due to their propensity to accumulate defects in DNA synthesis, making them more susceptible to a suboptimal M phase (e.g. triggered by ROCKi). However, if the source of SL was solely associated with DNA synthesis events, then it would also be present in BRCA1-deficient backgrounds, which we did not observe. Importantly, BRCA1-deficient backgrounds are also vulnerable during M phase, as we previously observed SL between BRCA1 and PLK1 inhibitors (*Carbajosa et al., 2019*). The identification of synthetic lethal interactions specific for BRCA1 or BRCA2, indicates that HR impairment is not the only possible trigger of SL in BRCA1- and BRCA2-deficient backgrounds that could be therapeutically exploited. In the future, M phase may provide a window of opportunity for novel treatments in patients that do not respond to PARPi therapy.

## Cytokinesis failure as the trigger of the SL between BRCA2-deficiency and ROCK inhibition

We believe that DNA replication defects are not the main trigger for the SL observed with ROCKi, and that defects intrinsic to M phase are more likely to account for ROCKi-induced cell death of BRCA2-deficient cells. Intriguingly, BRCA2 and ROCK functions converge at cytokinesis. ROCK kinases accumulate at the cleavage furrow (*Kosako et al., 2000*; *Yokoyama et al., 2005*), regulate furrow ingression, and their knockdown induces multinucleation (*Yokoyama et al., 2005*). Similarly, CITK localizes to the cleavage furrow, and its downregulation or inhibition also causes multinucleation (*Kosako et al., 2000*; *Sahin et al., 2019*). Cytokinesis defects have also been reported for BRCA2-deficient cells (*Daniels et al., 2004*; *Jonsdottir et al., 2009*; *Mondal et al., 2012*; *Rowley et al., 2011*). However, BRCA2 localizes to a different cytokinesis structure than ROCK, the midbody (*Daniels et al., 2004*; *Jonsdottir et al., 2009*; *Mondal et al., 2012*; *Rowley et al., 2011*). Remarkably, previous reports suggest that the effect of BRCA2 downregulation on cytokinesis regulation may be very mild (*Lekomtsev et al., 2010*). Given ROCK and BRCA2 localize to cytokinesis structures that are also separated in time (furrow and midbody), the deficiency in both functions may potentiate cytokinesis failure and cell death. Supporting cytokinesis failure as the SL trigger between ROCK and BRCA2 deficiency backgrounds, we observed that binucleation significantly accumulates at 24 h of treatment, when other mitotic defects have not yet significantly increased.

SL can be enhanced by the formation of multipolar spindles due to centrosome amplification. BRCA2 contributes to the regulation of centriole splitting (*Saladino et al., 2009*) and centrosome number (*Ehlén et al., 2020*; *Saladino et al., 2009*; *Tutt et al., 1999*). BRCA2 also localizes to centrosomes and preventing such a localization causes centrosome amplification and multinucleation (*Shailani et al., 2018*). ROCK also localizes to the centrosome (*Chevrier et al., 2002*; *Ma et al., 2006*) and its activity is required for centrosome movement and positioning (*Chevrier et al., 2002*; *Rosenblatt et al., 2004*). Similar to BRCA2 deficiency, ROCK inhibition also induces centriole splitting and centrosome amplification (*Aoki et al., 2009*; *Chevrier et al., 2002*; *Oku et al., 2014*). Interestingly, both ROCK and BRCA2 bind nucleophosmin (NPM/B23), a protein involved in the timely initiation of centrosome duplication (*Ma et al., 2006*; *Okuda et al., 2000*) and disrupting the interaction between BRCA2 and NPM/B23 induces centrosome fragmentation and multinucleation (*Wang et al., 2011*). Hence, the SL observed after BRCA2 deficiency and ROCKi may be enhanced by centrosome dysregulation, leading to mitotic spindle defects, cytokinesis failure and cell death. Further work may shed

additional light on this SL pathway and unravel other potential druggable targets that could provide therapeutic alternatives for treating BRCA2-deficient tumors.

## Materials and methods

### Screening

Stable HCT116$^{p21-/-}$ cell lines tagged with fluorescent proteins (CFP, iRFP or mCherry) and expressing Scramble, BRCA1, or BRCA2 shRNAs (*Carbajosa et al., 2019*) were co-cultured in equal proportions in 96-well plates for 6 days in the presence (0.1 µM) of each of the 680 compounds of the Protein Kinase Inhibitor Set 2 (PKIS2) library (*Drewry et al., 2017*; *Elkins et al., 2016*). At the end of treatment, the final cell number for each cell population was assessed with an automated flow cytometer Attune NxT acoustic focusing cytometer (Thermo Fisher). olaparib (#S1060, SelleckChem) at 100 nM was used as a positive control in each screening plate.

For each tested compound, two scenarios are possible: (A) non-selective effect, where the ratio of the populations remains unchanged. The non-selective compounds can either be non-toxic (the number of cells in all populations remains the same) or toxic (the number of cells from each population decreases similartestly); (B) synthetic lethal: selective toxicity against the BRCA2-deficient population, thus changing the relative abundance and ratio between the different populations. Additionally, a compound was considered a 'hit' if it exhibited a>5 standard deviation on two values: (1) Fold of SL induction, calculated from the ratios of the different populations in each well; and (2) Survival difference, calculated from the differential survival when comparing a given treatment to the untreated wells in the same plate. For more extensive details on the screening platform and calculations used for the analysis, please refer to *Carbajosa et al., 2019*.

### Lentiviral production

Lentiviral shRNA vectors were generated by cloning shBRCA2 (5′-AACTGAGCAAGCCTCAGTCA ACTCGAGTTGACTGAGGCTTGCTCAGTT) or shScramble (5′-GTTAACTGCGTACCTTGAGTA) into the pLKO.1-TRC vector (*Grotsky et al., 2013*). HEK293T cells were transfected with pLKO.1 and packaging plasmids (psPAX, and pMD2.G) 24 h post-seeding using JetPrime transfection reagent (Polyplus). After another 24 hr, media was changed. Forty-eight h after, media was collected, centrifuged, and supernatants were aliquoted and stored at –80 °C. Optimal viral titers were tested by serial dilutions and selected based on the minimal toxicity observed in the target cells.

### Generation of HCT116$^{p21-/-}$ shRNA stable cell lines

HCT116$^{p21-/-}$ cells (a kind gift from Bert Volgelstein, Johns Hopkins University) were used to generate stable shScramble or shBRCA2 HCT116$^{p21-/-}$ cells using lentiviral transduction. For viral transduction cells were seeded in 60 mm dishes, and 24 h post-seeding they were transduced using optimal viral titer and 8 µg/ml polybrene (#sc-134220, Santa Cruz Biotechnology). Transduced cells were selected with 1 µg/ml puromycin (#P8833, Sigma-Aldrich) 24 h post-transduction, and amplified for later freezing. Frozen stocks were not used for more than three weeks after thawing. BRCA2 knockdown was confirmed using quantitative real-time PCR.

### Other cell lines and culture conditions

PEO1/PEO4: PEO1 is a BRCA2-deficient ovarian cell line derived from the ascites fluid of a patient (*Langdon et al., 1988*; *Wolf et al., 1987*). PEO4 derives from the same patient after the development of chemotherapy resistance and BRCA2 function recovery (*Sakai et al., 2009*; *Wolf et al., 1987*). V-C8 and V-C8#13: V-C8 (a kind gift from Bernard Lopez, Gustave Roussy Cancer Center) is a BRCA2-deficient Chinese hamster lung cell line, while V-C8#13 has restored BRCA2 function via one copy of human chromosome 13 harboring BRCA2 (*Kraakman-van der Zwet et al., 2002*). DLD-1/ DLD-1$^{BRCA2-/-}$ cell lines (# HD PAR-008 and #HD 105–00, Horizon Discovery Ltd.): DLD-1 cell lines are human colorectal cancer cell lines, while the BRCA2-deficient DLD-1$^{BRCA2-/-}$ cell line has BRCA2 exon 11 disrupted with rAAV gene editing technology (*Hucl et al., 2008*).

PEO4/PEO1 and DLD-1/DLD-1$^{BRCA2-/-}$ cell lines were grown in RPMI (#31800–089, Gibco) supplemented with 10% fetal bovine serum (Natocor) and 1% penicillin/streptomycin. V-C8#13 /V-C8, HCC1937$^{BRCA1}$/HCC1937 (ATCC) and HEK293T (a kind gift from Alejandro Schinder, Fundación

Instituto Leloir) were grown in DMEM (#12800082, Gibco) supplemented with 10% fetal bovine serum (Natocor) and 1% penicillin/streptomycin. All cell lines were maintained in a humidified, 5% $CO_2$ incubator and passaged as needed. Cell lines were regularly checked for mycoplasma contamination. The BRCA2 and BRCA1 status of all cell lines was checked, and none of the used cell lines is in the list of commonly misidentified cell lines maintained by the International Cell Line Authentication Committee.

## Drugs and treatments

Cells were treated 24 h post-seeding. Treatment times for each experiment, ranging from 24 h to 6 days, are specified below or in the figure legends. olaparib (#S1060, SelleckChem) was resuspended in DMSO and stored at –20 °C. ROCK inhibitors, fasudil HCl (#A10381, Adooq), SR 3677 dihydrochloride (A12674) and ripasudil (#S7995, SelleckChem) were resuspended in water and stored at –80 °C. BrdU (Sigma-Aldrich) was resuspended in DMSO and stored at –20 °C. BrdU-containing media (10 µM) was added to cell cultures 15 min before harvest. Cisplatin was resuspended in 0.9% NaCl and stored at –20 °C (#P4394, Sigma-Aldrich). Cisplatin was added to cell cultures for 24 hr. All drug stocks were filter-sterilized (0.2 µM). Unless otherwise stated, all experiments were performed three times.

## Survival assay

To perform a survival assay that can be directly compared with the phenotypic screening used in this report we plated in each single well from a 96-well plate, a number of cells that would reach 90% confluence at the time of finalization of the assay (6 days). HCT116$^{p21-/-}$ cell lines were seeded at 1500 cells/well, V-C8 at 500 cells/well, PEO at 2500 cells/well and DLD-1/DLD-1$^{BRCA2-/-}$ at 500 and 1500 cells/well, respectively. Cells were treated with the indicated reagents 24 h post-seeding. Each treatment had three technical replicates. The last day, cells were fixed with 4% paraformaldehyde/ 2% sucrose and stained with DAPI (#10236276001, Roche). Plates were photographed with the IN Cell Analyzer 2200 high content analyzer (GE Healthcare), using a ×10 objective. A total of nine pictures per individual well were taken, and all nuclei in the image were automatically counted to assess cell numbers for each well. Cell number (%) after each treatment was calculated relative to the total number of cells in untreated wells in the same plate. In this way and similarly to the phenotypic screening, cells were counted directly and no indirect metabolic parameter, sub G1 populations or other parameters were monitored. Also, variables such as extreme dilutions (e.g.: used in clonogenic survival) were not introduced by this assay.

## Restriction enzyme digestion

Genomic DNA from PEO4 and PEO1 cell lines was extracted using phenol-chloroform-isoamyl alcohol (#P3803, Sigma-Aldrich). A fragment of 694 bp within the BRCA2 gene was PCR amplified using specific primers (Forward primer: AGATCACAGCTGCCCCAAAG, Reverse primer: TTGCGTTGAGGA ACTTGTGAC). PCR fragments were gel purified, and equal amounts of DNA were subject to DrdI (New England Biolabs) enzyme digestion following the manufacturer's instructions. Digestion products were run on an agarose gel and stained with ethidium bromide to visualize the band pattern.

## Chromosome aberration analysis

Cells were seeded and treated 24 h post-seeding, and 0.08 µg/ml colcemid (KaryoMAX, Invitrogen) was added 20 h before harvest. Following trypsinization, cell pellets were incubated in hypotonic buffer (KCl 0.0075 M) at 37 °C for 4 min and fixed with Carnoy's fixative solution (3:1 methanol: glacial acetic acid). Cells were dropped onto slides and air-dried before staining with 6% Giemsa in Sorensen's buffer (2:1 67 mM $KH_2PO_4$:67 mM $Na_2HPO_4$, pH 6.8) for 2 min. Pictures of metaphases were taken using an automated Applied Imaging Cytovision microscope (Leica Biosystems). Fifty metaphase spreads per independent experiment were analyzed for chromosome gaps, breaks and exchanges.

## Anaphase aberration assay

Cells were fixed with 2% paraformaldehyde/ 2% sucrose for 20 min and stained with DAPI (#10236276001, Roche) to visualize anaphases and quantify anaphase aberrations (bridges and lagging chromosomes). At least 50 anaphases/sample were analyzed. Z-stacks were acquired with a

Zeiss LSM 510 Meta confocal microscope and were combined for image generation. Maximum intensity projections were generated using FIJI (ImageJ) Imaging Software.

## Micronuclei assay

Micronuclei (MN) analyses were performed using protocols previously described by us (*Federico et al., 2016*). Briefly, cells were seeded at low density, treated and incubated with cytochalasin B (4.5 µg/ml, Sigma-Aldrich) for 40 hr. Cells were washed twice with PBS and fixed with PFA/sucrose 2% for 20 min. Phalloidin and DAPI staining were used to visualize whole cells and nuclei, respectively. A total of 300 binucleated cells were analyzed, and the frequency was calculated as MN/binucleated cells.

## Immunofluorescence

Cells were seeded on coverslips, treated, fixed for 20 min with 2% paraformaldehyde/ 2% sucrose and permeabilized for 15 min with 0.1% Triton-X 100. Following 1 h blocking with 2.5% donkey serum in 0.05% PBS/Tween, coverslips were incubated as needed with primary antibodies: γH2AX S139 (1:1500, #05–636-I, Millipore), 53BP1 (1:1500, #sc-22760, Santa Cruz Biotechnology), cyclin A (1:1000, #GTX-634–420, GeneTex) or Phalloidin (1:50, #A12379, Invitrogen). For BrdU staining (1:500, #RPN20AB, GE Healthcare), cells were fixed with ice-cold methanol (40 s) and acetone (20 s), followed by DNA denaturing in 1.5 N HCl for 40 min. For staining of centrosomes (1:1000, #T6557, Sigma-Aldrich) and microtubules (1:1000, #T9026, Sigma-Aldrich), cells were fixed for 10 min with ice-cold methanol, followed by hydration with PBS. Following 1 h of incubation with primary antibodies, cells were washed (3 x/10 min each) with 0.05% PBS/Tween, incubated for 1 h with anti-donkey Alexa 488 or 546 (1:200, Invitrogen), washed, stained with DAPI (#10236276001, Roche) and mounted on slides with Mowiol (Sigma-Aldrich). Slides were analyzed with ×40 or100 x objectives using an Axio Observer microscope (Zeiss).

## Number of 53BP1 foci

Cells were seeded on coverslips and treated as described in the immunofluorescence section above. The quantification of foci/cell was executed using the protocol used by *Kilgas et al., 2021*. For the experiment in *Figure 3—figure supplement 2*, in which cells were treated for 6 days, because of the presence of bi and multinucleation, the number of 53BP1 foci per cell was normalised according to their number of nuclei, resulting in the number of 53BP1 foci/nuclei informed.

## Colony Assay

shScramble and shBRCA2 HCT116$^{p21-/-}$ cells were treated with fasudil for 24 hr. Samples were washed and the cells attached to the plate were trypsinized, counted and seeded at extremely low density in 24-well plates. After 10–12 days of culture, the media was removed, and crystal violet staining solution was added for colony visualization. The crystal violet staining solution was washed with ddH$_2$O. The colony assay was performed utilizing for different cell dilutions. The cell colony number was determined as described in *Joray et al., 2017*.

## Flow cytometry analysis

Cells were seeded, treated and harvested at different time points (24 hr-6 days). For propidium iodide staining, cells were trypsinized, fixed with ice-cold ethanol overnight, and stained with a solution of 100 µg/ml RNase (#10109142001, Roche) and 50 µg/ml propidium iodide (#P4170, Sigma-Aldrich). A total of 10,000 events were recorded using a FACSCalibur (BD Biosciences). Cell cycle distribution was analyzed with the Cytomation Summit software (Dako version 4.3). To assess cell death using SYTOX Green, cells were treated and harvested at different time points. Following trypsinization, samples were stained with SYTOX Green staining following manufacturer's instructions (#S34860, Invitrogen). 10,000 events were recorded and analyzed using a FACSAria (BD Biosciences).

## Quantitative real-time PCR

Total RNA was extracted with TRIzol reagent (Invitrogen), following the manufacturer's instructions. A total of 2 µg of RNA was used as a template for cDNA synthesis using M-MLV reverse transcriptase (#28025, Invitrogen) and oligo-dT as primer. Quantitative real-time PCR was performed in a

LightCycler 480 II (Roche) using the 5 X HOT FIREPol EvaGreen q PCR Mix Plus (#08-24-00001, Solis BioDyne).

To calculate relative expression levels, samples were normalized to GAPDH expression. Forward (FW) and reverse (RV) primers were as follows: BRCA2 (FW: AGGGCCACTTTCAAGAGACA, RV:TAGT TGGGGTGGACCACTTG), ROCK1 (FW: GATATGGCTGGAAGAAACAGTA, RV:TCAGCTCTATAC ACATCTCCTT), ROCK2 (FW:AGATTATAGCACCTTGCAAAGTA, RV:TATCTTTTTCACCAACCGAC TAA), CITK (FW:CAGGCAAGATTGAGAACG, RV:GCACGATTGAGACAGGGA), EMI1 (FW:TGTT CAGAAATCAGCAGCCCAG, RV:CAGGTTGCCCGTTGTAAATAGC) and GAPDH (FW:AGCCTCCC GCTTCGCTCTCT, RV GAGCGATGTGGCTCGGCTGG).

## siRNA transfection

siRNAs were transfected using JetPrime transfection reagent (Polyplus) following the manufacturer's instructions. Unless otherwise stated, cells were transfected for a total of 48 hr. siROCK1 (#sc-29473 Santa Cruz Biotechnology) and siROCK2 (#sc-29474, Santa Cruz Biotechnology) were used at 100 nM. siEMI1 (#sc-37611 Santa Cruz Biotechnology) and siCITK (#sc-39214 Santa Cruz Biotechnology) were both used at 100 nM.

Scale Bar: Scales bars were automatically calculated using the Image J program. *Figure 2A–D* 100 µm, *Figure 3B* 10 µm, *Figure 3E* 10 µm, *Figure 4D* 10 µm, *Figure 5A* 10 µm, *Figure 6A* 10 µm, *Figure 6D* 8 µm, *Figure 8B* 100 µm, *Figure 8C* 10 µm, *Figure 8E* 100 µm, *Figure 3—figure supplement 1A* 10µm, *Figure 3—figure supplement 1B* 10 µm, *Figure 5—figure supplement 1B* 10 µm, *Figure 6—figure supplement 1C* 10 µm, *Figure 8—figure supplement 1A* 100 µm, *Figure 8—figure supplement 1B* 100 µm.

## Statistical analysis

GraphPad Prism 5.0 was used for all statistical analyses. Regular two-way ANOVA, followed by a Bonferroni post-test or Student's t-tests were used as appropriate. BrdU intensity was analyzed with a Kruskal-Wallis non-parametric test followed by a Dunn's multiple comparison test. Statistical significance was set at $p < 0.05$.

## Acknowledgements

We thank Dr. Fernanda Ledda for providing critical reagents for this work. We would also like to thank all Gottifredi and Soria Laboratories members for their insightful comments and discussions. We thank Pamela Rodriguez, Esteban Miglietta, Andrés Hugo Rossi and Carla Pascuale for technical support with tissue culture, microscopy and flow cytometry. We also thank the flow cytometry, microscopy, and cell culture facilities of CIBICI-CONICET for technical support. This work was supported by a consortium grant of FONCyT and the Trust in Science Program (Global Health R&D) from GlaxoSmithKline (PAE-GLAXO 2014–0005) to JLB and (PCE-GSK 2017–0032) to GS and PICT 2018–01857 and L'Oréal-UNESCO National Award 2019 to VG. JLB, GS and VG are researchers from the National Council of Scientific and Technological Research (CONICET). JM, SOS, NSP and SC were supported by fellowships from the National Agency for the Promotion of Science and Technology (ANPCyT). SOS, NLC, NSP, CG and MFP were supported by fellowships from CONICET. MFP was supported by a fellowship from the National Institute of Cancer (Argentina).

## Additional information

### Competing interests

Maria F Pansa, Israel Gloger, Gerard Drewes, Kevin P Madauss: is affiliated with GlaxoSmithKline and has no other competing interests to declare. The other authors declare that no competing interests exist.

## Funding

| Funder | Grant reference number | Author |
| --- | --- | --- |
| L'Oréal | National Award 2019 | Vanesa Gottifredi |
| Agencia Nacional de Promoción Científica y Tecnológica | PAE-GLAXO 2014–0005 | José Luis Bocco |
| GlaxoSmithKline | PAE-GLAXO 2014–0005 | José Luis Bocco |
| Agencia Nacional de Promoción Científica y Tecnológica | PCE-GSK 2017–0032 | Gastón Soria |
| GlaxoSmithKline | PCE-GSK 2017–0032 | Gastón Soria |
| Agencia Nacional de Promoción Científica y Tecnológica | PICT 2018–0185 | Vanesa Gottifredi |
| Consejo Nacional de Investigaciones Científicas y Técnicas | Researcher | José Luis Bocco Gastón Soria Vanesa Gottifredi |
| Agencia Nacional de Promoción Científica y Tecnológica | Fellowship | Julieta Martino Sebastián Omar Siri Natalia Soledad Paviolo Sofía Carbajosa |
| Consejo Nacional de Investigaciones Científicas y Técnicas | Fellowship | Sebastián Omar Siri Nicolás Luis Calzetta Natalia Soledad Paviolo Cintia Garro Maria F Pansa |
| Instituto Nacional del Cáncer | Fellowship | Maria F Pansa |

The funders had no role in study design, data collection and interpretation, or the decision to submit the work for publication.

## Author contributions

Julieta Martino, Data curation, Formal analysis, Validation, Investigation, Visualization, Methodology, Writing – original draft, Writing – review and editing; Sebastián Omar Siri, Data curation, Formal analysis, Validation, Investigation, Visualization, Methodology, Writing – review and editing; Nicolás Luis Calzetta, Data curation, Formal analysis, Validation, Investigation, Visualization, Methodology; Natalia Soledad Paviolo, Data curation, Formal analysis, Validation, Investigation, Visualization; Cintia Garro, Data curation, Formal analysis, Investigation, Visualization; Maria F Pansa, Sofía Carbajosa, Israel Gloger, Gerard Drewes, Data curation, Formal analysis; Aaron C Brown, Data curation, Formal analysis, Methodology; José Luis Bocco, Resources, Funding acquisition; Kevin P Madauss, Data curation, Formal analysis, Supervision, Methodology; Gastón Soria, Conceptualization, Resources, Data curation, Supervision, Funding acquisition, Investigation, Methodology, Project administration, Writing – review and editing; Vanesa Gottifredi, Conceptualization, Resources, Supervision, Funding acquisition, Investigation, Methodology, Writing – original draft, Project administration, Writing – review and editing

## Author ORCIDs

Sebastián Omar Siri ![ORCID] http://orcid.org/0000-0002-0945-5605
José Luis Bocco ![ORCID] http://orcid.org/0000-0002-9682-1270
Vanesa Gottifredi ![ORCID] http://orcid.org/0000-0001-9656-5951

## Decision letter and Author response

Decision letter https://doi.org/10.7554/eLife.80254.sa1
Author response https://doi.org/10.7554/eLife.80254.sa2

## Additional files

### Supplementary files
- MDAR checklist

### Data availability
All data generated or analysed during this study are included in the manuscript and supporting file.

The following dataset was generated:

| Author(s) | Year | Dataset title | Dataset URL | Database and Identifier |
|---|---|---|---|---|
| Martino J, Siri S, Paviolo NS, Garro C, Pansa M, Carbajosa S, Brown A, Bocco J, Gloger I, Drewes G, Madauss K, Soria G, Gottifredi V | 2022 | Phenotypic screening assay on co-culture BRCA-proficient and BRCA-deficient cell lines | https://dx.doi.org/10.5061/dryad.ht76hdrjj | Dryad Digital Repository, 10.5061/dryad.ht76hdrjj |

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
