## [Editor Report]

This paper reports the fundamental discovery that BRCA2-deficient cells are highly sensitive to the inhibition or depletion of Rho-kinases (ROCK), known to regulate actin cytoskeleton dynamics. This observed synthetic lethality between ROCK and BRCA2 is suggested to be independent of acute replication stress, is outside of the cellular S phase and may represent a promising new synthetic lethality target for the treatment of BRCA2-deficient tumors.

---

## [Decision Letter]

**Decision letter after peer review:**

Thank you for submitting your article "Inhibitors of ROCK kinases induce multiple mitotic defects and synthetic lethality in BRCA2-deficient cells" for consideration by *eLife*. Your article has been reviewed by 3 peer reviewers, one of whom is a member of our Board of Reviewing Editors, and the evaluation has been overseen Wafik El-Deiry as the Senior Editor. The reviewers have opted to remain anonymous.

Essential revisions:

1) The evidence that this SL interaction is independent of replication defects is not solid. Please provide more data to support this conclusion.

2) The SL interaction is based on chemical inhibitors only, with 6 out of 9 ROCK inhibitors not demonstrating the SL interaction. Please explain and define why?

3) The mechanisms by which ROCKi specifically affects BRCA2-defective cells are elusive. Explain.

4) It remains unclear what the cause of the multiple mitotic defects is – best if the authors expand on this.

5) Missing is some discussion, at least, on how BRCA2 expression is related to actin cytoskeleton dynamics controlled by ROCK kinases.

6) The authors also show that ROCK kinase inhibitors are also quite toxic to WT cells. Is this via the same mechanism but just attenuated and if so, why?

7) The authors should provide more detail on the mechanism of BRCA2-specific cell death by ROCK kinase inhibitors

*Reviewer #1 (Recommendations for the authors):*

Authors focus all the PARPi effects in BRCA2-KO cells as due to HR defects, but recent reports have supported more clarity of that phenotype, related to the accumulation of replication gaps and not solely due to HR defects. Would be good to enhance the Intro and discussion accordingly.

Missing is some discussion, at least, on how BRCA2 expression is related to actin cytoskeleton dynamics controlled by ROCK kinases.

The authors point out how important these findings are and how it was dependent on their "novel phenotypic survival screening method". This should be explained in more detail. This is particularly important since the PE01 cells, using this method, are not very sensitive to PARPi's as expected.

The authors also show that ROCK kinase inhibitors are also quite toxic to WT cells. Is this via the same mechanism but just attenuated and if so, why?

The authors should provide more detail on the mechanism of BRCA2-specific cell death by ROCK kinase inhibitors.

Also, why do many of the ROCK kinase inhibitors evaluated show no phenotype?

*Reviewer #2 (Recommendations for the authors):*

– The SL interaction that the authors describe is potentially interesting and appears to be independent of replication stress. longer assays (ie clonogenic assays) and preferably in vivo work are required to demonstrate the value of this SL interaction.

– The evidence that this SL interaction is independent of replication defects is not solid. the observation depends on a few markers and could be strengthened by functional replication analysis (DNA fiber analysis). the cut-off for replication stress analysis is highly arbitrary (5 foci for 53BP1 and 35 foci for γH2AX). Absolute foci numbers should be plotted and statistically analyzed, especially because this is a key argument in the manuscript.

– The work depends solely on chemical inhibitors (of which 3 out of 9 score in the SL analysis). Why do the other ROCK inhibitors not show SL effects? Genetic approaches in long-term assays are required to strengthen the SL interaction.

– EME1 depletion allows cells to skip mitosis, but also restores HR in BRCA-deficient cells (work from Pagano lab). This effect should be considered.

– The observed G2/M arrest upon ROCKi could represent 4n G1 cells.

– The nature and cause of the observed anaphase bridges are enigmatic. A defect in cytokinesis does not explain the defects observed earlier in mitosis.

– It is unclear why ROCK or citron kinase selectively disrupt cytokinesis in BRCA2-defective cells? Also, depmap analysis shows that the viability of most cell lines is negatively affected by ROCK1 loss. How do the authors imagine that long-term loss of ROCK specifically affects BRCA2-defective cells? Is there evidence from public data that this interaction is BRCA2-specific?

– Aberrant multipolar spindles are quantified from DNA staining, but not spindle stainings. This requires analysis of spindle components.

– The chromosome bridge in figure 5A (left panel) does not seem to be a real bridge, but a chromosome that segregates late.

*Reviewer #3 (Recommendations for the authors):*

1. The timeline in the left panel of Figure 1D seems unnecessary. In addition, the axis should be abellin for the examples in the two right panels of Figure 1D.

2. Axis in Figure 2E is too small to read.

3. There are several formatting inconsistencies, typos, and grammatical errors throughout the manuscript and figures that should be carefully reviewed and corrected.

4. Figure 1 —figure supplement 1: The data shown is not overly convincing. Why wasn’t the [fasudil] taken out as far as it was in some of the BRCA2 deficient lines?

5. It would benefit readers less familiar with looking at metaphase spreads if the authors added arrows pointing out the aberrant phenotypic features.

6. In figure 6 the shades of gray and blue used are too similar. The black borders separating the colors get lost in the error bars especially when adjacent to the darkest blue/gray shades.

[Editors’ note: further revisions were suggested prior to acceptance, as described below.]

Thank you for resubmitting your work entitled “Inhibitors of Rho kinases (ROCK) induce multiple mitotic defects and synthetic lethality in BRCA2-deficient cells” for further consideration by *eLife*. Your revised article has been evaluated by Wafik El-Deiry (Senior Editor) and a Reviewing Editor.

The manuscript has been improved but there are some remaining issues that need to be addressed, as outlined below:

In particular, I would ask that you work to address the concerns raised by Reviewer #2 below.

*Reviewer #2 (Recommendations for the authors):*

The authors have addressed my main comments, although some major shortcomings remain.

– The main shortcoming is that there is no mechanistic data that explains why ROCK kinase inhibition selectively affects BRCA2-defective cells. The model that the authors put forward is that ROCK inhibition leads to multinucleate cells, selectively in BRCA2-depleted cells and that this leads in the next cell cycle to structural mitotic defects (eg lagging chromosomes, multipolar spindles). Although the analysis of early and late time points supports this model, any mechanistic underpinning is lacking. Why does ROCK inhibition selectively induce binucleated cells in BRCA2-defective cells in the first cell cycle? There is some literature cited, but that only provides a shallow explanation. Time-lapse imaging would be very informative here to confirm the model that is put forward.

– The analysis of numbers/intensity of H2AX/53BP1 foci in absolute measures are much more insightful than the arbitrary cut-off. I would suggest showing these data in the main figure not supplement.

– Line 123: one ->ones.

– Supplement Figure 3E: please check abelling: should be HU with the images of fibers?

– Clonogenic assays: this is substandard. Please also include the controls and dose response.

*Reviewer #3 (Recommendations for the authors):*

The manuscript is much improved by the revisions and I feel all of the essential revisions have been adequately addressed.

[Editors’ note: further revisions were suggested prior to acceptance, as described below.]

Thank you for resubmitting your work entitled “Inhibitors of Rho kinases (ROCK) induce multiple mitotic defects and synthetic lethality in BRCA2-deficient cells” for further consideration by *eLife*. Your revised article has been evaluated by Wafik El-Deiry (Senior Editor) and a Reviewing Editor.

The manuscript has been improved but there are some remaining issues that need to be addressed, as outlined below:

*Reviewer #2 (Recommendations for the authors):*

I appreciate that the authors have changed the manuscript, concerning data visualisation. Also, I value how the authors have changed their model and writing in which a first binucleation event precedes further toxic events, which aligns much better with the presented data.

I realize that I suggested live cell microscopy analysis, and that this comment came after the first revision. However, it is not per se a request for a specific technique, it was merely a request to provide some mechanistic insight. It was just a suggestion of a commonly used technique in the cell cycle/mitosis research field to uncover what the order of events is that happens in BRCA2-defective cells in which ROCK is inactivated.

Personally, I still feel that the mechanistic insight is not substantial and that the main novelty is that ROCK inactivation is more toxic in BRCA2-depleted cells when compared to BRCA2-proficient cells. I realize that the authors do not agree with this standpoint. At least make clear in the discussion that it is unclear what happens mechanistically.

As the reviewers suggested, I have considered the 4 points put forward in the rebuttal letter (especially points 1,3 and 4):

1. ROCKi can kill BRCA2-deficient cells in a manner that is independent of replication stress in S phase. Such a concept has been reinforced in the R1 version by adding fiber assays and including the quantification of H2AX suggested by reviewer 2.

True, there is no overt replication stress induced by acute ROCK inhibition that can explain the SL effects. The mode of action of PARP inhibitor is clearly different than those of ROCK inhibitors in this respect. However, whether replication stress is not involved in this form of SL is difficult to conclude.

The effects of 53Bp1, H2AX and fibers were done after 48h of treatment, whereas the toxic effects of ROCK inhibition occur much later, when binucleated cells have I. Since whole genome duplication upon mitotic failure is known to cause replication stress (see for instance Gemble et al., Nature, 2022). To really conclude that replication stress is not involved, analysis at a late time points should be done. Of note. this point of the author does not indicate an underlying mechanism of action, but suggests a mechanism that is not involved.

2. The cytokinesis defects triggered by ROCKi have been documented by others, and so was the contribution of BRCA2-depletion to cytokinesis defects. By monitoring cells with DNA content higher than 4 N and binucleation, this manuscript shows that the treatment of BRCA2-deficient cells with ROCKi causes a significant augmentation of binucleation at early time points hence suggesting that the cytokinesis defects is a common defect that is amplified by the combined depletion of BRCA2 and the inhibition of ROCKi. Moreover, supporting the idea of cytokinesis defect as the trigger of other defects, including the DNAreplication independent SL, it temporally precedes all other mitotic defects.

The mechanism proposed here is that ROCKi and BRCA2 inactivation both give cytokinesis defects, and that the SL effect is based on these effects adding up. This is not entirely novel, and also, BRCA2 inactivation in this manuscript only marginally induces cytokinesis defects on its own. That cytokinesis failure leads to secondary defects I completely support, but claiming that secondarily effects are DNA replication independent is not shown and should not be claimed.

3. Reinforcing the link between cytokinesis failure and SL, the inhibition of another kinase that has also been associated with cytokinesis defect, Citron Rho-interacting kinase, also triggers SL in BRCA-2 deficient cells.

True, the observation that Rho kinase and ROCK behave the same is strong, and points towards cytokinesis, in which both kinases are clearly implicated.

4. The prevention of mitosis by EMI1 depletion prevents the SL caused by ROCKi in BRCA2- deficient cells, thus reinforcing the association between M-phase transit and cell death in BRCA2-deficient cells.

True, the EMI experiment connects mitosis to the SL effect of ROCK inhibition in these cells, reinforcing that a cytokinesis failure is involved

---

## [Author Response]

Essential revisions:1) The evidence that this SL interaction is independent of replication defects is not solid. Please provide more data to support this conclusion.

In the original version of the manuscript, we included the analysis of BrdU+ cells (%) and BrdU intensity in cells transiting S phase (current Figure 3D and Figure 3—figure supplement 1F). We believe that such evidence is crucial since BrdU intensity reveals global changes in the replication choreography that might be overlooked in DNA spreading assays (Calzetta, Gonzalez Besteiro, and Gottifredi, 2021). Moreover, such a negligible effect of ROCKi in the S phase progression of BRCA2 deficient cells is not associated with changes in γH2AX or 53BP1 nuclear foci accumulation (current Figure 3A-C).

Nonetheless, we addressed the reviewers' request to provide additional data by performing two sets of experiments:

1) Reviewer 2 specifically highlighted the need for an alternative quantification method of γH2AX or 53BP1 nuclear foci accumulation independent of arbitrary thresholds. Hence, we quantified absolute γH2AX intensity and the number of 53BP1 foci/ cell (current Supplementary Figure 3 A-D). Such analyses are shown in Figure 3—figure supplement 1AD, and the conclusions (replication stress levels are not upregulated in BRCA2-deficient cells treated with ROCKi) are aligned with the data reported in Figure 3A-C.2) We have also carried out a fiber assay analysis to monitor nascent DNA elongation tracks and origin firing frequency. ROCKi causes no changes in such DNA replication parameters in both BRCA2-WT and BRCA2-deficient cells (current Figure 3E-F). In contrast, hydroxyurea (HU) caused changes in the nascent DNA track length of BRCA2-deficient cells, as previously shown elsewhere (Schlacher et al., 2011) (current Figure 3—figure supplement 1E). Together, these results demonstrate that while BRCA2 deficiency alters DNA replication parameters after HU, it does not trigger obvious replication defects after the ROCK inhibitor, fasudil.

Hence, the data in Figure 3E-F and Figure 3—figure supplement 1A-E further support our claim of a SL interaction independent of replication defects.

2) The SL interaction is based on chemical inhibitors only, with 6 out of 9 ROCK inhibitors not demonstrating the SL interaction. Please explain and define why?

The SL interaction identified herein is not based on chemical inhibitors only. In fact, in the original version of this manuscript, we have provided genetic evidence (using siRNAs against ROCK1 and ROCK2) to confirm this SL interaction (current Figure 8B and Figure 8—figure supplement 1A).

When it comes to the reason for obtaining SL-positive results for only 3 out of 9 ROCK inhibitors in the screening, it is related to the low dosage at which the screening was performed. Given that ATP competitive kinase inhibitors are well known to hit multiple off-target kinases with similar ATP binding pockets, we decided to perform this screening at a low dose: 0.1 μM. As such, we aim to reduce false positives and the number of hits to validate, yet concomitantly risk increasing the rate of false negatives. This occurrence of false negatives was precisely the case with the ROCK inhibitors. While the 3 hits were within their activity range at the sub-micro-molar range (Figure 1 E and F), the 6 remaining inhibitors were out of their activity range. When re-testing the remaining 6 inhibitors a 1μM, they all showed SL activity on BRCA2-deficient cells. This information is now in Figure 1figure supplement 1.

In this revised version, we also included another potent and selective ROCK inhibitor (SR 3677 dihydrochloride) revealing SL activity in BRCA2-deficient samples (current Figure 2—figure supplement 2D).

3) The mechanisms by which ROCKi specifically affects BRCA2-defective cells are elusive. Explain.

The first type of mitotic defects we observe in BRCA2-deficient cells after fasudil treatment is a significant increase in polyploidy and high levels of binucleated cells (current Figure 4 A-C and Figure 7), which indicates that cytokinesis failure may be the trigger to the SL caused by ROCKi in BRCA2-deficient cells. As detailed below, both ROCK and BRCA2 (but not BRCA1) have been implicated in regulating cytokinesis during mitosis.

Evidence linking ROCK with cytokinesis regulation: ROCK accumulate at the cleavage furrow and phosphorylate myosin II, a component of the actomyosin ring whose contraction gives rise to the cleavage furrow which divides the cytoplasm (Kosako et al., 2000; Yokoyama, Goto, Izawa, Mizutani, and Inagaki, 2005). Depending on the cell line, inhibiting ROCK can impair or delay furrow ingression (Kosako et al., 2000; Lordier et al., 2008). If furrowing is impaired, cytokinesis fails, and there is polyploidization (Lordier et al., 2008). Knockdown of ROCK1 or ROCK2 using siRNAs also induces multinucleation (Yokoyama et al., 2005).

Evidence linking BRCA2 with cytokinesis regulation: BRCA2 localizes to a different cytokinesis structure than ROCK, the midbody, where it mediates the assembly of factors involved in abscission (Daniels, Wang, Lee, and Venkitaraman, 2004; Jonsdottir et al., 2009); (Mondal et al., 2012; Rowley et al., 2011). Cytokinesis defects have also been reported for BRCA2-deficient cell models (Daniels et al., 2004; Jonsdottir et al., 2009; Lee, Daniels, Garnett, and Venkitaraman, 2011; Rowley et al., 2011; Vinciguerra, Godinho, Parmar, Pellman,’and D'Andrea, 2010) (Mondal et al., 2012). Importantly, this bona fide role of BRCA2 in cytokinesis is independent of its role in HR (Mondal et al., 2012).

The evidence above indicates that both BRCA2 and ROCK regulate cytokinesis, albeit at different levels of the process. Such contributions must be mild or easy to compensate for, as individual deficiencies in our experimental settings are insufficient to impair cytokinesis. On this note, previous reports suggest that the individual effects may be very mild. For example, (Lekomtsev, Guizetti, Pozniakovsky, Gerlich, and Petronczki, 2010) compared BRCA2 knockdown against the knockdown of MgcRacGAP, a well-known mediator of cytokinesis, and the multinucleation levels were comparably so low that they concluded that BRCA2 had no significant contribution to cytokinesis.

Our results indicate that the initial trigger that precedes the mitotic defects observed when ROCK inhibition is combined with BRCA2-deficient backgrounds is binucleation (detected as early as 24 hours post-treatment-Figure 7 A). Given that ROCK and BRCA2 localize to cytokinesis structures (furrow and midbody) at different time points, we propose that the combined deficiencies may cause defects in the early and late steps of cytokinesis which augments the probability of cytokinesis failure.

4) It remains unclear what the cause of the multiple mitotic defe–ts is – best if the authors expand on this.

In BRCA2-deficient cells treated with ROCKi, the cytokinesis failure is accompanied by other mitotic defects that, we believe, derive from the DNA replication proficiency maintained by binucleated BRCA2-deficient cells (current Figure 6—figure supplement 1C-D). Because multipolar mitosis events accumulate at later time points than binucleation (48 and 24 hours of treatment respectively current Figure 7), we speculate that the attempt to separate DNA after multipolar spindle assembly could ultimately trigger cell death in BRCA2-deficient cells treated with fasudil. We also detected increased centrosome numbers in such samples (current Figure 6D). Centrosomes, the organelles responsible for nucleating the mitotic spindle, must be in the correct number (two) for successful chromosome segregation during mitosis (Fukasawa, 2007). Increased centrosome numbers (i.e., centrosome amplification) lead to abnormal spindle poles, chromosome instability or cell death (Ganem, Godinho, and Pellman, 2009). Centrosome amplification can result from multiple mechanisms, including cytokinesis failure (Fukasawa, 2007; Lens and Medema, 2019). Centrosome duplication is also ruled by timely centriole splitting (Tsou and Stearns, 2006), and in BRCA2.deficient tumors high levels of centriole splitting have been observed (Saladino, Bourke, Conroy, and Morrison, 2009). BRCA2 also localizes to centrosomes, and preventing this localization causes centrosome amplification and multinucleation (Han, Saito, Miki, and Nakanishi, 2008; Nakanishi et al., 2007; Zhang et al., 2016). Consistent with in vitro results, BRCA2-deficient tumors display increased centrosome numbers (Hou, Li, Ren, and Liu, 2016; Saladino et al., 2009; Tutt et al., 1999; Watanabe et al., 2018; Wiegant, Overmeer, Godthelp, van Buul, and Zdzienicka, 2006). Additionally, BRCA2 has also been implicated in facilitating chromosome alignment and as part of the spindle assembly checkpoint (SAC), which regulates the metaphase to anaphase transition by monitoring proper kinetochoremicrotubules attachment (Choi et al., 2012; Ehlén et al., 2020). The loss of M phase functions in BRCA2-deficient cells, including the regulation of centrosome numbers, may synergistically interact with ROCKi to kill cells. ROCK also localizes to the centrosome (Chevrier et al., 2002; Ma et al., 2006), and its activity is required for centrosome movement and positioning (Chevrier et al., 2002; Rosenblatt, Cramer, Baum, and McGee, 2004), which are crucial to proper spindle formation as well as mitotic exit (Piel, Nordberg, Euteneur, and Bornens, 2001). Inhibition of ROCK causes mitotic spindle misorientation, diffuse pericentrin and increases the nucleating capacity of astral microtubules (Heng et al., 2012; Rosenblatt et al., 2004). Like BRCA2 deficiency, ROCK inhibition also induces centriole splitting and centrosome amplification (Aoki, Ueda, Kataoka, and Satoh, 2009; Chevrier et al., 2002; Oku et al., 2014). Interestingly, both ROCK and BRCA2 bind nucleophosmin (NPM/B23), a protein involved in the timely initiation of centrosome duplication (Ma et al., 2006; Okuda et al., 2000) and disrupting the interaction between BRCA2 and NPM/B23 induces centrosome fragmentation and multinucleation (Wang, Takenaka, Nakanishi, and Miki, 2011).

Together, these findings may explain why BRCA2 deficiency and ROCKi potentiate the accumulation of multiple mitotic defects.

5) Missing is some discussion, at least, on how BRCA2 expression is related to actin cytoskeleton dynamics controlled by ROCK kinases.

As discussed in points 3 and 4, cytokinesis is affected by BRCA2 loss and ROCKi at different levels; while BRCA2 loss dysregulates the midbody organization, ROCKi impair the mitotic furrow formation. Additionally, ROCK and BRCA2 localize to the centrosomes and prevent their amplification. While BRCA2 prevents dysregulation of centriole number and participates in the SAC, ROCK regulates the movement and positioning of centrosomes. Hence, the SL does not result from BRCA2 controlling the ROCK-mediated cytoskeleton regulation. Instead, ROCK and BRCA2 have complementing functions that favor cell fitness in mitosis.

6) The authors also show that ROCK kinase inhibitors are also quite toxic to WT cells. Is this via the same mechanism but just attenuated and if so, why?

We designed an experiment to test whether WT cells die via the same mechanism by which ROCKi kills BRCA2-deficient cells. To test this, we increased fasudil doses in WT cells to achieve levels of cell death similar to those obtained after lower doses of fasudil in BRCA2-deficient cells. Surprisingly, despite the massive cell death observed in WT cells at high fasudil concentrations, increased binucleation was not observed (current Figure 7—figure supplement 1). Hence, the treatment of BRCA2-deficient cells with ROCKi triggers a cell death preceded by cytokinesis failure, while the treatment of WT cells with higher doses of ROCKi triggers a cell death independent of cytokinesis failure.

7) The authors should provide more detail on the mechanism of BRCA2-specific cell death by ROCK kinase inhibitors.

The evidence discussed in points 3 and 4 and the experiment reported in point 6 have already provided additional details on the BRCA2-specific cell death by ROCKi. Furthermore, based on one of the comments from reviewer 2: "The nature and cause of the observed anaphase bridges are enigmatic; a defect in cytokinesis does not explain the defects observed earlier in mitosis", we designed an experiment that provides further detail to the SL mechanism. We monitored binucleation, anaphase and mitotic aberrations at early times: 24, 48 and 72 hours after fasudil treatment. As shown in Figure 7, binucleated cells but not anaphase and mitotic aberrations, were detected at 24 hours in BRCA2-deficient cells treated with ROCKi. These data support the model in Figure 8G, indicating that the earliest alterations that accumulate in BRCA2-deficient cells treated with ROCKi are binucleated cells. Given that DNA duplication is not inhibited in binucleated cells (current Figure 6—figure supplement 1C-D), it is very likely that the anaphase bridges do not precede cytokinesis but accumulate later when binucleated or multinucleated cells attempt mitosis. From these data, we infer that the attempt to achieve separation of polyploid nuclei into daughter cells is the most likely trigger of cell death in BRCA2-deficient cells treated with ROCKi.

Reviewer #1 (Recommendations for the authors):Authors focus all the PARPi effects in BRCA2-KO cells as due to HR defects, but recent reports have supported more clarity of that phenotype, related to the accumulation of replication gaps and not solely due to HR defects. Would be good to enhance the Intro and discussion accordingly.

Thank you very much for this comment. Reviewer 1 is correct; the introduction has been improved, and the bibliography has been updated. Please see changes in Page 4 of the introduction section.

Missing is some discussion, at least, on how BRCA2 expression is related to actin cytoskeleton dynamics controlled by ROCK kinases.

This comment is part of the "essential revision" section. Please read responses to points 3-5 above.

The authors point out how important these findings are and how it was dependent on their "novel phenotypic survival screening method". This should be explained in more detail. This is particularly important since the PE01 cells, using this method, are not very sensitive to PARPi's as expected.

The novel method we refer to is the screening method described by Carbajosa et al., 2019. Because of reviewer 1's comment, we added details to the automatized cell counting methodology used for PEO1/PEO4 analysis (page 17; Materials and methods).

The authors also show that ROCK kinase inhibitors are also quite toxic to WT cells. Is this via the same mechanism but just attenuated and if so, why?

This comment is part of the "essential revision" section. Please read point 6 from that section.

The authors should provide more detail on the mechanism of BRCA2-specific cell death by ROCK kinase inhibitors

This comment is part of the "essential revision" section. Please read point 7 from that section.

Also, why do many of the ROCK kinase inhibitors evaluated show no phenotype?

This comment is part of the "essential revision" section. Please read point 2 from that section.

Reviewer #2 (Recommendations for the authors):– The SL interaction that the authors describe is potentially interesting and appears to be independent of replication stress. longer assays (ie clonogenic assays) and preferably in vivo work are required to demonstrate the value of this SL interaction.

This comment is part of the "essential revision" section. Please read point 1 from that section. We have performed clonogenic assays (see current Figure 2—figure supplement 1). We could not set up an in vivo model in the time dedicated to this revision.

– The evidence that this SL interaction is independent of replication defects is not solid. the observation depends on a few markers and could be strengthened by functional replication analysis (DNA fiber analysis). the cut-off for replication stress analysis is highly arbitrary (5 foci for 53BP1 and 35 foci for γH2AX). Absolute foci numbers should be plotted and statistically analyzed, especially because this is a key argument in the manuscript.

The arbitrary cuts have been established based on the replication stress induced by PARP inhibitors in BRCA2-deficient cells, but the comments from reviewer 2 are appropriate, as the replication stress induced by another agent could differ from the one caused by PARP inhibitors. As requested by reviewer 2, absolute γH2AX intensity and 53BP1 foci number have been analyzed and reported in Figure 3—figure supplement 1A-D. Our conclusion has not changed as both markers increased in PARPi-treated but not ROCKi-treated samples.

– The work depends solely on chemical inhibitors (of which 3 out of 9 score in the SL analysis). Why do the other ROCK inhibitors not show SL effects? Genetic approaches in long-term assays are required to strengthen the SL interaction.

This comment is part of the "essential revision" section. Please refer to point 2 from that section.

– EME1 depletion allows cells to skip mitosis, but also restores HR in BRCA-deficient cells (work from Pagano lab). This effect should be considered.

Thank you for bringing up this manuscript about the role of EMI1 in controlling RAD51 protein levels in human breast cancer samples, which reveals the role of EMI1 in HR. While the regulation of RAD51 by EMI1 is undoubtedly relevant when interpreting the effect of EMI1 on PARPi-induced synthetic lethality, some observations from our manuscript do not support a potential role of HR in the SL we describe. First, there is no replication stress in BRCA2-deficient cells treated with ROCKi, so the accumulation of double-strand breaks is unlikely. Second, ROCKi is synthetic lethal in BRCA2- but not BRCA1-deficient cells, and a SL involving HR modulation should equally affect both backgrounds. Third, ROCK overlap with BRCA2 in controlling mitotic events at the level of cytokinesis and centriole regulation (see points 3 and 4 in the "essential revision" section), but ROCK have no role in HR.

– The observed G2/M arrest upon ROCKi could represent 4n G1 cells.

Thank you for spotting this possibility that could explain the early accumulation of cells in the 4n region of the flow cytometry peaks. We have indicated it in the manuscript (see page 8, in the Results section).

– The nature and cause of the observed anaphase bridges are enigmatic. a defect in cytokinesis does not explain the defects observed earlier in mitosis.

This comment is part of the "essential revision" section. Please see point 6 from that section.

– It is unclear why ROCK or citron kinase selectively disrupt cytokinesis in BRCA2-defective cells? Also, depmap analysis shows that the viability of most cell lines is negatively affected by ROCK1 loss. How do the authors imagine that long-term loss of ROCK specifically affects BRCA2-defective cells? Is there evidence from public data that this interaction is BRCA2-specific?

Citron Kinase is intimately related to the function of ROCK, and how ROCK and BRCA2 defects can enhance cytokinesis failure have been described in the "essential revision" section (point 3). Regarding the DepMap analysis, we are not sure which data the reviewer is referring to, as ROCK1 knockout by CRISPR was essential only in 15 of the 1086 cell lines studied, and knockdown by RNAi did not reveal essentiality phenotypes in any of the 710 cell lines tested (https://depmap.org/portal/gene/ROCK1?tab=overview).

Regarding evidence from public databases regarding a specific interaction between BRCA2 and ROCK1, we couldn't find any previous link reported.

– Aberrant multipolar spindles are quantified from DNA staining, but not spindle stainings. This requires analysis of spindle components.

Our first indication of multipolar spindles were aberrant metaphases observed in experiments done only with DAPI staining (Figure 4 D-E). This was followed up by adding staining against α-tubulin (microtubules) and γ-tubulin (centrosomes) (Figure 6 D-F). Such an information is reported in the Methods section and in the legend of Figure 6. The quantifications in Figure 6E and 6F relied on spindle staining (revealed by γ- tubulin), centrosomes (revealed by alfa- tubulin) and DNA (revealed by DAPI). As shown in Figure 6 D, only anaphases revealing all three immunofluorescent marks were considered in the analysis (and all metaphases without exception were positive for the three marks). We realize that the label “percent of cells” may be misleading and we changed it to “Percent of metaphases”.

– The chromosome bridge in figure 5A (left panel) does not seem to be a real bridge, but a chromosome that segregates late.

We have selected a better representative image which is now part of Figure 6 A.

Reviewer #3 (Recommendations for the authors):1. The timeline in the left panel of Figure 1D seems unnecessary. In addition, the axis should be labeled for the examples in the two right panels of Figure 1D.

We have introduced the changes suggested by the reviewer. Thank you.

2. Axis in Figure 2E is too small to read.

The size of the font was enlarged, as suggested by reviewer 3.

3. There are several formatting inconsistencies, typos, and grammatical errors throughout the manuscript and figures that should be carefully reviewed and corrected.

We have carefully proofread the manuscript.

4. Figure 1 —figure supplement 1: The data shown is not overly convincing. Why wasn't the [Fasudil] taken out as far as it was in some of the BRCA2 deficient lines?

Fasudil curves were the same in BRCA1- and BRCA2-deficient samples such as HCT116 (Figure 2 A for BRCA2-deficient vs WT samples and Supplementary Figure 1K for BRCA1-deficient vs WTt samples). We could not reach higher doses in the BRCA1-deficient sample because the control sample (present in both curves) was already affected by the highest dose. Further support to the absence of SL by fasudil in BRCA1-deficient samples is in Figure 2—figure supplement 2E. Please note that at higher doses, the control samples are even more sensitive than the BRCA1-deficient samples to fasudil (the opposite of what we observed in BRCA2-deficient samples).

5. It would benefit readers less familiar with looking at metaphase spreads if the authors added arrows pointing out the aberrant phenotypic features.

We have added arrows to the aberrant phenotypic feature in Figure 5 A.

6. In figure 6 the shades of gray and blue used are too similar. The black borders separating the colors get lost in the error bars especially when adjacent to the darkest blue/gray shades.

We have modified the colors to make the figure easier to understand. Thank you.

[Editors' note: further revisions were suggested prior to acceptance, as described below.]

Reviewer #2 (Recommendations for the authors):The authors have addressed my main comments, although some major shortcomings remain.– The main shortcoming is that there is no mechanistic data that explains why ROCK kinase inhibition selectively affects BRCA2-defective cells. the model that the authors put forward is that ROCK inhibition leads to multinucleate cells, selectively in BRCA2-depleted cells and that this leads in the next cell cycle to structural mitotic defects (eg lagging chromosomes, multipolar spindles). although the analysis of early and late time points supports this model, any mechanistic underpinning is lacking. Why does ROCK inhibition selectively induce binucleated cells in BRCA2-defective cells in the first cell cycle? there is some literature cited, but that only provides a shallow explanation. time-lapse imaging would be very informative here to confirm the model that is put forward.

We submitted the original version of this manuscript on June 2022 and provided an R1 version of the manuscript attending all reviewer comments in November 2022. The live cell microscopy experiment has been suggested by reviewer 2 when evaluating the R1 version of our manuscript. We do not deny that such an experiment could be very interesting to perform as many others but we cannot provide this experiment at this time. It is technically challenging and impossible to perform without the building up of expertise that can only be gained after several months.

Moreover, we believe that the comment indicating that “no mechanistic data that explains why ROCK kinase inhibition selectively affects BRCA2-defective cells” is a bit harsh. As discussed below, there are many critical mechanistic insides which are already provided by our manuscript (when assessing the mechanistic contribution reviewer 2 focuses on point 2 but not 1, 3 and 4). We ask reviewer 2 and the editors to please consider the following mechanistic insights provided in the manuscript:

1) ROCKi can kill BRCA2-deficient cells in a manner that is independent of replication stress in S phase. Such a concept has been reinforced in the R1 version by adding fiber assays and including the quantification of H2AX suggested by reviewer 2.2) The cytokinesis defects triggered by ROCKi have been documented by others, and so was the contribution of BRCA2-depletion to cytokinesis defects. By monitoring cells with DNA content higher than 4 N and binucleation, this manuscript shows that the treatment of BRCA2-deficient cells with ROCKi causes a significant augmentation of binucleation at early time points hence suggesting that the cytokinesis defects is a common defect that is amplified by the combined depletion of BRCA2 and the inhibition of ROCKi. Moreover, supporting the idea of cytokinesis defect as the trigger of other defects, including the DNA replication independent SL, it temporally precedes all other mitotic defects.3) Reinforcing the link between cytokinesis failure and SL, the inhibition of another kinase that has also been associated with cytokinesis defect, Citron Rho-interacting kinase, also triggers SL in BRCA-2 deficient cells.4) The prevention of mitosis by EMI1 depletion prevents the SL caused by ROCKi in BRCA2deficient cells, thus reinforcing the association between M-phase transit and cell death in BRCA2-deficient cells.

Finally, reviewer 2 also considers that the analysis of early and late time points supports a model indicating that ROCKi first leads BRCA2-depleted cells to binucleation and later to structural mitotic defects (e.g. lagging chromosomes, multipolar spindles). In the manuscript, we have made it clear that our findings indicate that a significant increase in binucleation is detected earlier than other mitotic defects without mentioning the first or second cycle. In addition, in the current R2 version of the manuscript, we have changed a sentence “we speculate that after a subsequent DNA duplication, these cells can attempt mitosis.” (see lane 1075).

– The analysis of numbers/intensity of H2AX/53BP1 foci in absolute measures are much more insightful than the arbitrary cut-off. I would suggest showing these data in the main figure not supplement.

We have modified the text and Figures as suggested by reviewer 2.

– Line 123: one ->ones.

We have corrected such a typo.

– Supplement Figure 3E: please check labeling: should be HU with the images of fibers?

Reviewer 2 is right. Thank you for spotting the mistake.

– Clonogenic assays: this is substandard. please also include the controls and dose response.

We did not perform the experiment as a dose curve because the fasudil dose was selected from the survival assays. However, we performed the colony assay using different dilutions of cells. We are including all the analysis as well as the controls as requested by reviewer 2.

[Editors' note: further revisions were suggested prior to acceptance, as described below.]

Reviewer #2 (Recommendations for the authors):I appreciate that the authors have changed the manuscript, concerning data visualisation. Also, I value how the authors have changed their model and writing in which a first binucleation event precedes further toxic events, which aligns much better with the presented data.

We thank reviewer 2 for this positive assessment.

I realize that I suggested live cell microscopy analysis, and that this comment came after the first revision. However, it is not per se a request for a specific technique, it was merely a request to provide some mechanistic insight. It was just a suggestion of a commonly used technique in the cell cycle/mitosis research field to uncover what the order of events is that happens in BRCA2-defective cells in which ROCK is inactivated.Personally, I still feel that the mechanistic insight is not substantial and that the main novelty is that ROCK inactivation is more toxic in BRCA2-depleted cells when compared to BRCA2-proficient cells. I realize that the authors do not agree with this standpoint. At least make clear in the discussion that it is unclear what happens mechanistically.

We still believe that finding a SL association between a non-DDR kinase and BRCA2, which is independent from the induction of acute replication stress, is initiated by augmentation of binucleation, is prevented by EMI1 depletion and recapitulated by Citron Rho-Interacting kinase (another non-DDR kinase), and provides a certain level of mechanistic insight. We understand that reviewer 2 wants us to acknowledge that we have not identified the molecular target of ROCKi that triggers the excess cytokinesis failure in BRCA2-deficient cells. We have made this limitation explicit in the Discussion section of the R3 version of this manuscript (lines 328- 332).

As the reviewers suggested, I have considered the 4 points put forward in the rebuttal letter (especially points 1,3 and 4):1. ROCKi can kill BRCA2-deficient cells in a manner that is independent of replication stress in S phase. Such a concept has been reinforced in the R1 version by adding fiber assays and including the quantification of H2AX suggested by reviewer 2.True, there is no overt replication stress induced by acute ROCK inhibition that can explain the SL effects. The mode of action of PARP inhibitor is clearly different than those of ROCK inhibitors in this respect. However, whether replication stress is not involved in this form of SL is difficult to conclude.The effects of 53Bp1, H2AX and fibers were done after 48h of treatment, whereas the toxic effects of ROCK inhibition occur much later, when binucleated cells have occured. Since whole genome duplication upon mitotic failure is known to cause replication stress (see for instance Gemble et al., Nature, 2022). To really conclude that replication stress is not involved, analysis at a late time points should be done. Of note. this point of the author does not indicate an underlying mechanism of action, but suggests a mechanism that is not involved.

Because this section starts with “As the reviewers suggested…” we assume that this part of the letter has been written by the reviewing editor or the editor. We would like to emphasize that we never claimed that the replication stress was absent. We focused on acute replication stress which is the type of stress triggered by PARP inhibitors. Our intention was to highlight the mechanistic differences between ROCK and PARP inhibitors. However, and due to this request, we have performed the experiment requested herein by reviewer 2 and the editors. Surprisingly, there is no replication stress at later time points for mono, bi and multinucleated cells, which reinforce the replication stress-independent nature of this SL between ROCKi and BRCA2deficiency.

2. The cytokinesis defects triggered by ROCKi have been documented by others, and so was the contribution of BRCA2-depletion to cytokinesis defects. By monitoring cells with DNA content higher than 4 N and binucleation, this manuscript shows that the treatment of BRCA2-deficient cells with ROCKi causes a significant augmentation of binucleation at early time points hence suggesting that the cytokinesis defects is a common defect that is amplified by the combined depletion of BRCA2 and the inhibition of ROCKi. Moreover, supporting the idea of cytokinesis defect as the trigger of other defects, including the DNAreplication independent SL, it temporally precedes all other mitotic defects.The mechanism proposed here is that ROCKi and BRCA2 inactivation both give cytokinesis defects, and that the SL effect is based on these effects adding up. This is not entirely novel, and also, BRCA2 inactivation in this manuscript only marginally induces cytokinesis defects on its own. That cytokinesis failure leads to secondary defects I completely support, but claiming that secondarily effects are DNA replication independent is not shown and should not be claimed.

We did not claim independence from replication stress. Throughout the whole manuscript (original, R1 and R2) we did purposely limit such a conclusion to acute replication stress which is what we tested. Because of the experiments requested by the editors and reviewer 2 in this R3 round, we have monitored replication stress at the same time we measure cell death (6 days) by means of H2AX intensity and 53BP1 foci formation, in two cell lines (shBRCA2 HCT116^p21-/-^ and PEO1 BRCA2-/-) (see new Figure 3—figure supplement 2). Surprisingly, there was no replication stress even at those late time points for mono, bi and multinucleated cells. Such observation reinforces the dissociation of this SL from replication stress.

3. Reinforcing the link between cytokinesis failure and SL, the inhibition of another kinase that has also been associated with cytokinesis defect, Citron Rho-interacting kinase, also triggers SL in BRCA-2 deficient cells.True, the observation that Rho kinase and ROCK behave the same is strong, and points towards cytokinesis, in which both kinases are clearly implicated.

Thank you for agreeing with point 3.

4. The prevention of mitosis by EMI1 depletion prevents the SL caused by ROCKi in BRCA2- deficient cells, thus reinforcing the association between M-phase transit and cell death in BRCA2-deficient cells.True, the EMI experiment connects mitosis to the SL effect of ROCK inhibition in these cells, reinforcing that a cytokinesis failure is involved.

Thank you for agreeing with point 4.